# FIXING ASYMPTOTIC UNCERTAINTY OF BAYESIAN NEURAL NETWORKS WITH INFINITE ReLU FEATURES

## ABSTRACT

Approximate Bayesian methods can mitigate overconfidence in ReLU networks. However, far away from the training data, even Bayesian neural networks (BNNs) can still underestimate uncertainty and thus be overconfident. We suggest to fix this by considering an infinite number of ReLU features over the input domain that are never part of the training process and thus remain at prior values. Perhaps surprisingly, we show that this model leads to a tractable Gaussian process (GP) term that can be added to a pre-trained BNN's posterior at test time with negligible cost overhead. The BNN then yields structured uncertainty in the proximity of training data, while the GP prior calibrates uncertainty far away from them. As a key contribution, we prove that the added uncertainty yields cubic predictive variance growth, and thus the ideal uniform (maximum entropy) confidence in multi-class classification far from the training data.

## 1 INTRODUCTION

Calibrated uncertainty is crucial for safety-critical decision making by neural networks (NNs) (Amodei et al., 2016). Standard training methods of NNs yield point estimates that, even if they are highly accurate, can still be severely overconfident (Guo et al., 2017). Approximate Bayesian methods, which turn NNs into Bayesian neural networks (BNNs), can be used to address this issue. Kristiadi et al. (2020) recently showed that for binary ReLU classification networks, far away from the training data (more precisely: when scaling any input $x$ with a scalar $\alpha > 0$ and taking the limit $\alpha \to \infty$), the uncertainty of BNNs can be bounded away from zero. This is an encouraging result when put in contrast to the standard point-estimated networks, for which Hein et al. (2019) showed earlier that the same asymptotic limit always yields arbitrarily high (over-)confidence. Nevertheless, BNNs can still be asymptotically overconfident (albeit less so than the standard NNs) since the aforementioned uncertainty bound can be loose. This issue is our principal interest in this paper. An intuitive interpretation is that ReLU NNs "miss out on some uncertainty" even in their Bayesian formulation, because they fit a finite number of ReLU features to the training data, by "moving around" these features within the coverage of the data. This process has no means to encode a desideratum that the model should be increasingly uncertain away from the data.

In this work, we "add in" additional uncertainty by considering an infinite number of additional ReLU features spaced at regular intervals away from the data in the input and hidden spaces of the network. Since these features have negligible values in the data region, they do not contribute to the training process. Hence, we can consider a prior for their weights, chosen to be an independent Gaussian, and arrive at a specific Gaussian process (GP) which covariance function is a generalization of the classic cubic-spline kernel (Wahba, 1990). This GP prior can be added to *any* pre-trained ReLU BNN as a simple augmentation to its output. Considering the additive combination of a parametric BNN and GP prior together, we arrive at another view of the method: It approximates the "full GP posterior" that models the residual of a point-estimated NN (Blight & Ott, 1975; Qiu et al., 2020). In our factorization, the BNN models uncertainty *around* the training data, while the GP prior models uncertainty *far away* from them. By factorizing these two parts from each other, our formulation requires no (costly) GP posterior inference, and thus offers lightweight, modular uncertainty calibration. See Fig. 1 for illustration.

Theoretical analysis is a core contribution of this work. We show that the proposed method (i) preserves the predictive performance of the base ReLU BNN. Furthermore, it (ii) ensures that the

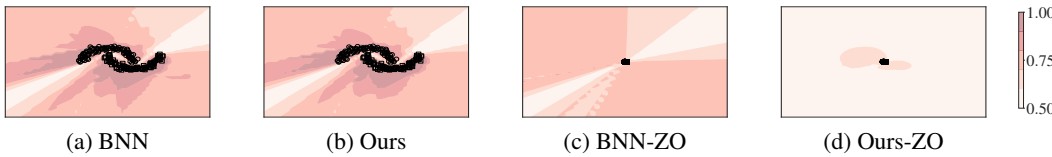

(a) BNN        (b) Ours        (c) BNN-ZO        (d) Ours-ZO

Figure 1: Toy classification with a BNN and our method. Shade represents confidence, the suffix "ZO" stands for "zoomed-out". Far away from the training data, vanilla BNNs can still be overconfident **(a, c)**. Our method fixes this issue while keeping predictions unchanged **(b, d)**.

surrounding output variance asymptotically grows cubically in the distance to the training data, and thus (iii) yields uniform asymptotic confidence in the multi-class classification setting. These results extend those of Kristiadi et al. (2020) in so far as their analysis is limited to the binary classification case and their bound can be loose. Furthermore, our approach is complementary to the method of Meinke & Hein (2020) which attains maximum uncertainty far from the data for non-Bayesian point-estimate NNs. Finally, our empirical evaluation confirms our analysis and shows that the proposed method also improves uncertainty estimates in the non-asymptotic regime.

## 2 BACKGROUND

### 2.1 BAYESIAN NEURAL NETWORKS

Let $f : \mathbb{R}^N \times \mathbb{R}^D \to \mathbb{R}^C$ defined by $(\boldsymbol{x}, \boldsymbol{\theta}) \mapsto f(\boldsymbol{x}; \boldsymbol{\theta}) =: f_{\boldsymbol{\theta}}(\boldsymbol{x})$ be a neural network. Here, $\boldsymbol{\theta}$ is the collection of all parameters of $f$. Given an i.i.d. dataset $\mathcal{D} := (\boldsymbol{x}_m, y_m)_{m=1}^M$, the standard training procedure amounts to finding a point estimate $\boldsymbol{\theta}^*$ of the parameters $\boldsymbol{\theta}$, which can be identified in the Bayesian framework with ***maximum a posteriori (MAP) estimation***[1]

$$\boldsymbol{\theta}^* = \arg\max_{\boldsymbol{\theta}} \log p(\boldsymbol{\theta} \mid \mathcal{D}) = \arg\max_{\boldsymbol{\theta}} \sum_{m=1}^M \log p(y_m \mid f_{\boldsymbol{\theta}}(\boldsymbol{x}_m)) + \log p(\boldsymbol{\theta}).$$

While this point estimate may yield highly accurate predictions, it does not encode uncertainty over $\boldsymbol{\theta}$, causing an overconfidence problem (Hein et al., 2019). Bayesian methods can mitigate this issue, specifically, by treating the parameter of $f$ as a random variable and applying Bayes rule. The resulting network is called a ***Bayesian neural network (BNN)***. The common way to approximate the posterior $p(\boldsymbol{\theta} \mid \mathcal{D})$ of a BNN is by a Gaussian $q(\boldsymbol{\theta} \mid \mathcal{D}) = \mathcal{N}(\boldsymbol{\theta} \mid \boldsymbol{\mu}, \boldsymbol{\Sigma})$, which can be constructed for example by a Laplace approximation (MacKay, 1992b) or variational Bayes (Hinton & Van Camp, 1993). Given such an approximate posterior $q(\boldsymbol{\theta} \mid \mathcal{D})$ and a test point $\boldsymbol{x}_* \in \mathbb{R}^N$, one then needs to marginalize the parameters to make predictions, i.e. we compute the integral $y_* = \int h(f(\boldsymbol{x}_*; \boldsymbol{\theta})) \, q(\boldsymbol{\theta} \mid \mathcal{D}) \, d\boldsymbol{\theta}$, where $h$ is an inverse link function, such as the identity function for regression or the logistic-sigmoid and softmax functions for binary and multi-class classifications, respectively. Since the network $f$ is a non-linear function of $\boldsymbol{\theta}$, this integral does not have an analytic solution. However, one can obtain a useful approximation via the following ***network linearization***: Let $\boldsymbol{x}_* \in \mathbb{R}^N$ be a test point and $q(\boldsymbol{\theta} \mid \mathcal{D}) = \mathcal{N}(\boldsymbol{\theta} \mid \boldsymbol{\mu}, \boldsymbol{\Sigma})$ be a Gaussian approximate posterior. Linearizing $f$ around $\boldsymbol{\mu}$ yields the following *marginal* distribution over the function output $f(\boldsymbol{x}_*)$:[2]

$$p(f(\boldsymbol{x}_*) \mid \boldsymbol{x}_*, \mathcal{D}) \approx \mathcal{N}(f(\boldsymbol{x}_*) \mid \underbrace{f(\boldsymbol{x}_*; \boldsymbol{\mu})}_{=:\boldsymbol{m}_*}, \underbrace{\boldsymbol{J}_*^\top \boldsymbol{\Sigma} \boldsymbol{J}_*}_{=:\boldsymbol{V}_*}), \tag{1}$$

where $\boldsymbol{J}_*$ is the Jacobian of $f(\boldsymbol{x}_*; \boldsymbol{\theta})$ w.r.t. $\boldsymbol{\theta}$ at $\boldsymbol{\mu}$. (In the case of a real-valued network $f$, we use the gradient $\boldsymbol{g}_* := \nabla_{\boldsymbol{\theta}} f(\boldsymbol{x}_*; \boldsymbol{\theta})|_{\boldsymbol{\mu}}$ instead of $\boldsymbol{J}_*$.) This distribution can then be used as the predictive distribution $p(y_* \mid \boldsymbol{x}_*, \mathcal{D})$ in the regression case. For classifications, we need another approximation since $h$ is not the identity function. One such approximation is the ***generalized probit approximation***

---

[1]In the statistical learning view, $\log p(y_m \mid f_{\boldsymbol{\theta}}(\boldsymbol{x}_m))$ is identified with the empirical risk, $\log p(\boldsymbol{\theta})$ with the regularizer. The two views are equivalent in this regard.

[2]See Bishop (2006, Sec. 5.7.1) for more details.

(Gibbs, 1997; Spiegelhalter & Lauritzen, 1990; MacKay, 1992a):

$$p(y_* = c \mid \boldsymbol{x}_*, \mathcal{D}) \approx \frac{\exp(m_{*c}\,\kappa_{*c})}{\sum_{i=1}^{C} \exp(m_{*i}\,\kappa_{*i})}, \qquad \text{for all } c = 1, \dots, C, \qquad (2)$$

where for each $i = 1, \dots, C$, the real numbers $m_{*i}$ is the $i$-th component of the vector $\boldsymbol{m}_*$, and $\kappa_{*i} := (1 + \pi/8\,v_{*ii})^{-1/2}$ where $v_{*ii}$ is the $i$-th diagonal term of the matrix $\boldsymbol{V}_*$. These approximations are analytically useful, but can be expensive due to the computation of the Jacobian matrix $\boldsymbol{J}_*$. Thus, ***Monte Carlo (MC) integration*** is commonly used as an alternative, i.e. we approximate $y_* \approx \frac{1}{S}\sum_{s=1}^{S} h(f(\boldsymbol{x}_*; \boldsymbol{\theta}_s))$ with $\boldsymbol{\theta}_s \sim q(\boldsymbol{\theta} \mid \mathcal{D})$. Finally, given a classification predictive distribution $p(y_* \mid \boldsymbol{x}_*, \mathcal{D})$, we define the predictive ***confidence*** of $\boldsymbol{x}_*$ as the maximum probability $\mathrm{conf}(\boldsymbol{x}_*) := \max_{c \in \{1,\dots,C\}} p(y_* = c \mid \boldsymbol{x}_*, \mathcal{D})$ over class labels.

## 2.2 ReLU and Gaussian processes

The ReLU activation function $\mathrm{ReLU}(z) := \max(0, z)$ (Nair & Hinton, 2010) has become the *de-facto* choice of non-linearity in deep learning. Given arbitrary real numbers $c$, it can be generalized as $\mathrm{ReLU}(z; c) := \max(0, z - c)$, with the "kink" at location $c$. An alternative formulation, useful below, is in terms of the Heaviside function $H$ as $\mathrm{ReLU}(z; c) = H(z - c)(z - c)$. We may define a collection of $d$ such ReLU functions evaluated at some point in $\mathbb{R}$ as the function $\boldsymbol{\phi} : \mathbb{R} \to \mathbb{R}^K$ with $z \mapsto (\mathrm{ReLU}(z; c_1), \dots, \mathrm{ReLU}(z; c_K))^\top$. We call this function the ***ReLU feature map***; it can be interpreted as "placing" ReLU functions at different locations in $\mathbb{R}$.

Consider a linear model $g : \mathbb{R} \times \mathbb{R}^K \to \mathbb{R}$ defined by $g(x; \boldsymbol{w}) := \boldsymbol{w}^\top \boldsymbol{\phi}(x)$. Suppose $\boldsymbol{\phi}$ regularly places the $K$ generalized ReLU functions centered at $(c_i)_{i=1}^K$ over $[c_{\min}, c_{\max}] \subset \mathbb{R}$, where $c_{\min} < c_{\max}$. If we consider a Gaussian prior $p(\boldsymbol{w}) := \mathcal{N}\left(\boldsymbol{w} \mid \boldsymbol{0}, \sigma^2 K^{-1}(c_{\max} - c_{\min})\boldsymbol{I}\right)$ over the weights $\boldsymbol{w}$ then, as $K$ goes to infinity, the distribution over $g(x)$ is a Gaussian process with mean $0$ and covariance (using the shorthand $g_x := g(x)$ and $\bar{x} := \min(x, x')$; full derivation in Appendix A):

$$\lim_{K \to \infty} \mathrm{cov}(g_x, g_{x'}) = \sigma^2 H(\bar{x} - c_{\min}) \left( \frac{1}{3}(\bar{x}^3 - c_{\min}^3) - \frac{1}{2}(\bar{x}^2 - c_{\min}^2)(x + x') + (\bar{x} - c_{\min})xx' \right)$$
$$=: k^1(x, x'; c_{\min}, \sigma^2),$$

for $\bar{x} \leq c_{\max}$. Since this expression does not depend on $c_{\max}$, we consider the limit $c_{\max} \to \infty$. The resulting covariance function is the ***cubic spline kernel*** (Wahba, 1990).

## 3 Method

Hein et al. (2019) showed that the confidence of point-estimated ReLU networks (i.e. feed-forward nets which use piecewise-affine activation functions and are linear in the output layer) approaches 1 with increasing distance from the training data. For binary classification, Kristiadi et al. (2020) showed that Gaussian-approximated ReLU BNNs $f$ instead approach a constant confidence bounded away from 1, but not necessarily close to the maximum uncertainty value of $1/2$. Thus, just being Bayesian as such does not fix overconfidence entirely. A close look at their proof suggests that the issue is a structural limitation of the deep model itself: for any input $\boldsymbol{x}_*$ and a sufficiently large scalar $\alpha$, both the mean and standard deviation of the output $f(\alpha \boldsymbol{x}_*)$ are linear functions of $\boldsymbol{x}_*$. Intuitively, this issue arises because the net only has finitely many ReLU features available to "explain" the data, and thus it "lacks" ReLU features for modeling uncertainty away from the data.

In this section, we will utilize the cubic spline kernel to construct a new kernel and method that, intuitively speaking, adds an infinite number ReLU features away from the data to pre-trained BNNs. This construction adds the "missing" ReLU features and endows BNNs with super-quadratic output variance growth, without affecting predictions. All proofs are in Appendix B.

### 3.1 The Double-Sided Cubic Spline Kernel

The cubic spline kernel constructed above is non-zero only on $(c_{\min}, \infty) \subset \mathbb{R}$. To make it suitable for modeling uncertainty in an unbounded domain, we set $c_{\min} = 0$ and obtain a kernel $k_\to^1(x, x'; \sigma^2) := k^1(x, x'; 0, \sigma^2)$ which is non-zero only on $(0, \infty)$. Doing an entirely analogous

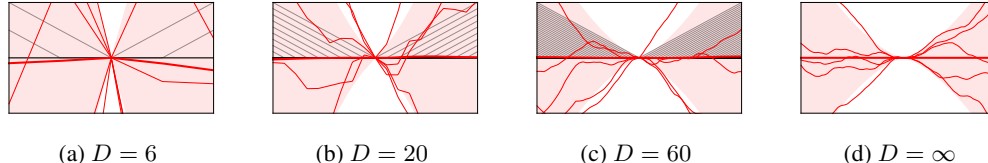

(a) $D = 6$      (b) $D = 20$      (c) $D = 60$      (d) $D = \infty$

Figure 2: The construction of our kernel in 1D, as the limiting covariance of the output of a Bayesian linear model with $D$ ReLU features. Grey curves are ReLU features while thin red curves are samples. Red shades are the $\pm 1$ standard deviations of those samples.

construction with infinitely many ReLU functions pointing to the left, i.e. $\mathrm{ReLU}(-z; c)$, we obtain the kernel $k^1_{\leftarrow}(x, x'; \sigma^2) := k^1_{\rightarrow}(-x, -x'; \sigma^2)$, which is non-zero only on $(-\infty, 0)$. We combine both into the kernel

$$k^1_{\leftrightarrow}(x, x'; \sigma^2) := k^1_{\leftarrow}(x, x'; \sigma^2) + k^1_{\rightarrow}(x, x'; \sigma^2),$$

which covers the whole real line (the value at the origin $k^1_{\leftrightarrow}(0, 0)$ is zero)—see Figure 2. For multivariate input domains, we define

$$k_{\leftrightarrow}(\boldsymbol{x}, \boldsymbol{x}'; \sigma^2) := \frac{1}{N} \sum_{i=1}^{N} k^1_{\leftrightarrow}(x_i, x'_i; \sigma^2) \tag{3}$$

for any $\boldsymbol{x}, \boldsymbol{x}' \in \mathbb{R}^N$ with $N > 1$. We here deliberately use a summation, instead of the alternative of a product, since we want the associated GP to add uncertainty anywhere where *at least* one input dimension has non-vanishing value.[3] We call this kernel the ***double-sided cubic spline (DSCS) kernel***. Two crucial properties of this kernel are that it has negligible values around the origin and for any $\boldsymbol{x}_* \in \mathbb{R}^N$ and $\alpha \in \mathbb{R}$, the value $k_{\leftrightarrow}(\alpha\boldsymbol{x}_*, \alpha\boldsymbol{x}_*)$ is *cubic* in $\alpha$.

### 3.2 ReLU-GP Residual

Let $f : \mathbb{R}^N \times \mathbb{R}^D \to \mathbb{R}$ be an $L$-layer, real-valued ReLU BNN. Suppose we place infinitely many ReLU features by following the previous construction. Then, we arrive at a zero-mean GP prior $\mathcal{GP}(\widehat{f}^{(0)} \mid 0, k_{\leftrightarrow})$ of some real-valued function $\widehat{f}^{(0)} : \mathbb{R}^N \to \mathbb{R}$ over the input space $\mathbb{R}^N$. We can use this GP to model the "missing" uncertainty which, due to the lack of its presence, makes $f$ overconfident far-away from the data. We do so in a standard manner by assuming that the "true" latent function $\widetilde{f}$ is the sum of $f$ and $\widehat{f}^{(0)}$:

$$\widetilde{f} := f + \widehat{f}^{(0)}, \qquad \text{where } \widehat{f}^{(0)} \sim \mathcal{GP}(\widehat{f}^{(0)} \mid 0, k_{\leftrightarrow}). \tag{4}$$

Under this assumption, given an input $\boldsymbol{x}_*$, it is clear that $\widehat{f}^{(0)}$ does not affect the expected output of the BNN since the GP over $\widehat{f}^{(0)}$ has zero mean. However, $\widehat{f}^{(0)}$ do additively affect the uncertainty of the BNN's output $f_* := f(\boldsymbol{x}_*)$ since if we assume that $f_* \sim \mathcal{N}(\mathbb{E} f_*, \mathrm{var} f_*)$, then it follows that $\widetilde{f}_* \sim \mathcal{N}(\mathbb{E} f_*, \mathrm{var} f_* + k_{\leftrightarrow}(\boldsymbol{x}_*, \boldsymbol{x}_*))$. Hence, the random function $\widehat{f}^{(0)}$, resulting from placing an infinite number of ReLU features in the input space, indeed models the *uncertainty residual* of the BNN $f$. We thus call our method ***ReLU-GP residual (RGPR)***.

Unlike previous methods for modeling residuals with GPs, RGPR does not require a posterior inference since intuitively, the additional infinitely many ReLU features are never part of the training process—their "kinks" are pointing away from the data. So even if we were to actively include them in the training process somehow, they would have (near) zero training gradient and just stay where and as they are. The following statements illustrate this intuition more formally in GP regression under the linearization (1) by assuming w.l.o.g. that the kernel values over the dataset are negligible (by shifting and scaling until the data is sufficiently close to $\mathbf{0} \in \mathbb{R}^N$).

---

[3]By contrast, a product $k_{\leftrightarrow}(\boldsymbol{x}, \boldsymbol{x}'; \sigma^2)$ is zero if one of the $k^1_{\leftrightarrow}(x_i, x'_i; \sigma^2)$ is zero.

**Proposition 1.** *Suppose $f : \mathbb{R}^N \times \mathbb{R}^D \to \mathbb{R}$ defined by $(\boldsymbol{x}, \boldsymbol{\theta}) \mapsto f(\boldsymbol{x}; \boldsymbol{\theta})$ is a ReLU regression BNN with a prior $p(\boldsymbol{\theta}) = \mathcal{N}(\boldsymbol{\theta} \mid \mathbf{0}, \boldsymbol{B})$ and $\mathcal{D} := \{\boldsymbol{x}_m, y_m\}_{m=1}^M$ is a dataset. Let $\widehat{f}^{(0)}$ and $\widetilde{f}$ be defined as in (4), and let $\boldsymbol{x}_* \in \mathbb{R}^N$ be arbitrary. Under the linearization of $f$ w.r.t. $\boldsymbol{\theta}$ around $\mathbf{0}$, given that all $\boldsymbol{x}_1, \dots, \boldsymbol{x}_M$ are sufficiently close to the origin, the GP posterior of $\widetilde{f}_* := \widetilde{f}(\boldsymbol{x}_*)$ is given by*

$$p(\widetilde{f}_* \mid \boldsymbol{x}_*, \mathcal{D}) \approx \mathcal{N}(\widetilde{f}_* \mid f(\boldsymbol{x}; \boldsymbol{\mu}), \boldsymbol{g}_*^\top \boldsymbol{\Sigma} \boldsymbol{g}_* + k_\leftrightarrow(\boldsymbol{x}_*, \boldsymbol{x}_*)), \tag{5}$$

*where $\boldsymbol{\mu}$ and $\boldsymbol{\Sigma}$ are the mean and covariance of the posterior of the linearized network, respectively, and $\boldsymbol{g}_* := \nabla_{\boldsymbol{\theta}} f(\boldsymbol{x}_*; \boldsymbol{\theta})|_{\mathbf{0}}$.*

The previous proposition shows that the GP prior of $\widehat{f}^{(0)}$ does not affect the BNN's approximate posterior—$\widetilde{f}$ is written as *a posteriori* $f$ plus *a priori* $\widehat{f}^{(0)}$. Therefore, given a pre-trained BNN $f$ with its associated posterior $p(\boldsymbol{\theta} \mid \mathcal{D}) \approx \mathcal{N}(\boldsymbol{\theta} \mid \boldsymbol{\mu}, \boldsymbol{\Sigma})$, we can simply add to its output $f(\boldsymbol{x}_*; \boldsymbol{\theta})$ (with $\boldsymbol{\theta} \sim p(\boldsymbol{\theta} \mid \mathcal{D})$) a random number $\widehat{f}^{(0)}(\boldsymbol{x}_*) \sim \mathcal{GP}(\widehat{f}^{(0)} \mid 0, k_\leftrightarrow(\boldsymbol{x}_*, \boldsymbol{x}_*))$. We henceforth assume that $f$ is a pre-trained BNN.

While the previous construction is sufficient for modeling uncertainty far away from the data, it does not model the uncertainty *near* the data region well. Figure 3(a) shows this behavior: placing infinitely many ReLU features over just the input space yields uncertainty that is not adapted to the data and hence, far away from them, we can still have low variance. To alleviate this issue, we additionally place infinite ReLU features on the representation space of the point-estimated $f_{\boldsymbol{\mu}}(\cdot) = f(\cdot; \boldsymbol{\mu})$, which indeed encodes information about the data since $f$ is a trained BNN, as follows.

For each $l = 1, \dots, L - 1$ and any input $\boldsymbol{x}_*$, let $N_l$ be the size of the $l$-th hidden layer of $f_{\boldsymbol{\mu}}$ and $\boldsymbol{h}^{(l)}(\boldsymbol{x}_*) =: \boldsymbol{h}_*^{(l)}$ be the $l$-th hidden units. By convention, we assume that $N_0 := N$ and $\boldsymbol{h}_*^{(0)} := \boldsymbol{x}_*$. We place for each $l = 0, \dots, L - 1$ an infinite number of ReLU features on the representation space $\mathbb{R}^{N_l}$, and thus we obtain a random function $\widehat{f}^{(l)} : \mathbb{R}^{N_l} \to \mathbb{R}$ distributed by the Gaussian process $\mathcal{GP}(\widehat{f}^{(l)} \mid 0, k_\leftrightarrow)$. Now, given that $\widehat{N} := \sum_{l=0}^{L-1} N_l$, we define the function $\widehat{f} : \mathbb{R}^{\widehat{N}} \to \mathbb{R}$ by $\widehat{f} := \widehat{f}^{(0)} + \dots + \widehat{f}^{(L-1)}$. This function is therefore a function over *all* representation (including the input) spaces of $f_{\boldsymbol{\mu}}$, distributed by the additive Gaussian process $\mathcal{GP}(\widehat{f} \mid 0, \sum_{l=0}^{L-1} k_\leftrightarrow)$. In other words, given the representations $\boldsymbol{h}_* := (\boldsymbol{h}_*^{(l)})_{l=0}^{L-1}$ of $\boldsymbol{x}_*$, the marginal over the function output $\widehat{f}(\boldsymbol{h}_*) =: \widehat{f}_*$ is thus given by

$$p(\widehat{f}_*) = \mathcal{N}\left(\widehat{f}_* \,\middle|\, 0, \sum_{l=0}^{L-1} k_\leftrightarrow\left(\boldsymbol{h}_*^{(l)}, \boldsymbol{h}_*^{(l)}; \sigma_l^2\right)\right). \tag{6}$$

Figure 3(c) visualizes the effect of this definition. The low-variance region modeled by the random function $\widehat{f}$ becomes more compact around the data and can be controlled by varying the kernel hyperparameter $\sigma_l^2$ for each layer $l = 0, \dots, L - 1$. Finally, we can then model the residual in (4) using $\widehat{f}$ instead, i.e. we assume $\widetilde{f} = f + \widehat{f}$.

The generalization of RGPR to BNNs with multiple outputs is straightforward. Let $f : \mathbb{R}^N \times \mathbb{R}^D \to \mathbb{R}^C$ be a vector-valued, pre-trained, $L$-layer ReLU BNN. We assume that the sequence of random functions $(\widehat{f}_c : \mathbb{R}^{\widehat{N}} \to \mathbb{R})_{c=1}^C$ is independent and identically distributed by the previous Gaussian process $\mathcal{GP}(\widehat{f} \mid 0, \sum_{l=0}^{L-1} k_\leftrightarrow)$. Thus, defining $\widehat{f}_* := \widehat{f}(\boldsymbol{h}_*) := (\widehat{f}_1(\boldsymbol{h}_*), \dots, \widehat{f}_C(\boldsymbol{h}_*))^\top$, we have

$$p(\widehat{f}_*) = \mathcal{N}\left(\widehat{f}_* \,\middle|\, \mathbf{0}, \sum_{l=0}^{L-1} k_\leftrightarrow\left(\boldsymbol{h}_*^{(l)}, \boldsymbol{h}_*^{(l)}; \sigma_l^2\right) \boldsymbol{I}\right). \tag{7}$$

Furthermore, as in the real-valued case, for any $\boldsymbol{x}_*$, the GP posterior $p(\widetilde{f}_* \mid \boldsymbol{x}_*, \mathcal{D})$ is approximately (under the linearization of $f$) given by the Gaussians derived from (1) and (7):

$$p(\widetilde{f}_* \mid \boldsymbol{x}_*, \mathcal{D}) \approx \mathcal{N}\left(\widetilde{f}_* \,\middle|\, f_{\boldsymbol{\mu}}(\boldsymbol{x}_*), \boldsymbol{J}_*^\top \boldsymbol{\Sigma} \boldsymbol{J}_* + \sum_{l=0}^{L-1} k_\leftrightarrow\left(\boldsymbol{h}_*^{(l)}, \boldsymbol{h}_*^{(l)}; \sigma_l^2\right) \boldsymbol{I}\right). \tag{8}$$

Although the derivations above may appear involved, it is worth emphasizing that in practice, the only overheads compared to the usual MC-integrated BNN prediction step are (i) a single additional forward-pass over $f_{\boldsymbol{\mu}}$, (ii) $L$ evaluations of the kernel $k_\leftrightarrow$ and (ii) sampling the $C$-dimensional Gaussian (7). Note that their costs are negligible compared to the cost of obtaining the standard MC-prediction of $f$. We refer the reader to Algorithm 1 for a step-by-step pseudocode.

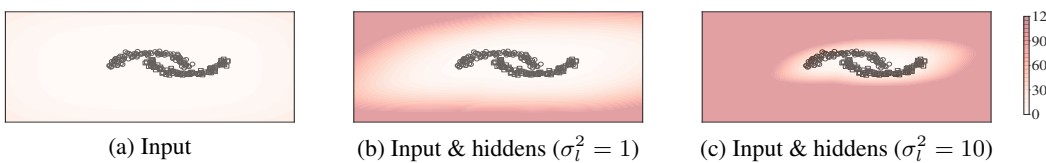

(a) Input      (b) Input & hiddens ($\sigma_l^2 = 1$)      (c) Input & hiddens ($\sigma_l^2 = 10$)

Figure 3: Variance of $\widehat{f}$ (6) as a function of $\boldsymbol{x}_*$. When $\widehat{f}$ is a function over neural network representations of the data (b), it captures the data region better than when $\widehat{f}$ is only defined on the input space (a). Increasing the kernel hyperparameters (here we assume they have the same value for all layers) makes the low-variance region more compact around the data (c).

---

**Algorithm 1** MC-prediction using RGPR. Differences from the standard procedure are in red.

**Input:**

    Pre-trained multi-class BNN classifier $f : \mathbb{R}^N \times \mathbb{R}^D \to \mathbb{R}^C$ with posterior $p(\boldsymbol{\theta} \mid \mathcal{D})$. Test point $\boldsymbol{x}_* \in \mathbb{R}^N$. Prior variance hyperparameters $(\sigma_l^2)_{l=0}^{L-1}$ of $\widehat{f}$. Inverse link function $h$. Number of MC samples $S$.

1:   $\{\boldsymbol{h}_*^{(l)}\}_{l=1}^{L-1} = \texttt{forward}(f_{\boldsymbol{\mu}}, \boldsymbol{x}_*)$      ▷ Compute representations of $\boldsymbol{x}_*$ via a forward pass on $f_{\boldsymbol{\mu}}$
2:   $v_s(\boldsymbol{x}_*) = \sum_{l=0}^{L-1} k_{\leftrightarrow}(\boldsymbol{h}_*^{(l)}, \boldsymbol{h}_*^{(l)}; \sigma_l^2)$      ▷ Compute the prior variance of $\widehat{f}$
3: **for** $s = 1, \dots, S$ **do**
4:     $\boldsymbol{\theta}_s \sim \mathcal{N}(\boldsymbol{\theta} \mid \boldsymbol{\mu}, \boldsymbol{\Sigma})$      ▷ Sample from the (approximate) posterior of $f$
5:     $\boldsymbol{f}_s(\boldsymbol{x}_*) = f(\boldsymbol{x}_*; \boldsymbol{\theta}_s)$      ▷ Forward pass on $f$ using the sampled parameter
6:     $\widehat{f}_s(\boldsymbol{x}_*) \sim \mathcal{N}(\widehat{f}(\boldsymbol{h}_*) \mid \boldsymbol{0}, v_s(\boldsymbol{x}_*)\boldsymbol{I})$      ▷ Sample from the marginal (7)
7:     $\widetilde{f}_s(\boldsymbol{x}_*) = f_s(\boldsymbol{x}_*) + \widehat{f}_s(\boldsymbol{x}_*)$      ▷ Compute $\widetilde{f}(\boldsymbol{x}_*; \boldsymbol{\theta}_s)$
8: **end for**
9: **return** $S^{-1} \sum_{s=1}^{S} h(\widetilde{f}_s(\boldsymbol{x}_*))$      ▷ Make prediction by averaging

---

## 4   ANALYSIS

Here, we will study the theoretical properties of RGPR. Our assumptions are mild: we (i) assume that RGPR is applied only to the input space and (ii) use the network linearization technique. Assumption (i) is the minimal condition for the results presented in this section to hold—similar results can also easily be obtained when hidden layers are also utilized in RGPR. Meanwhile, assumption (ii) is necessary for tractability—in Section 6 we will validate our analysis in general settings.

The following two propositions (i) summarize the property that RGPR preserves the original BNN's prediction and (ii) show that asymptotically, the marginal variance of the output of $\widetilde{f}$ grows cubically.

**Proposition 2 (Invariance in Predictions).** *Let $f : \mathbb{R}^N \times \mathbb{R}^D \to \mathbb{R}^C$ be any network with posterior $\mathcal{N}(\boldsymbol{\theta} \mid \boldsymbol{\mu}, \boldsymbol{\Sigma})$ and $\widetilde{f}$ be obtained from $f$ via RGPR (4). Then under the linearization of $f$, for any $\boldsymbol{x}_* \in \mathbb{R}^N$, we have $\mathbb{E}_{p(\widetilde{f}_* \mid \boldsymbol{x}_*, \mathcal{D})} \widetilde{f}_* = \mathbb{E}_{p(f_* \mid \boldsymbol{x}_*, \mathcal{D})} f_*$.*

**Proposition 3 (Asymptotic Variance Growth).** *Let $f : \mathbb{R}^N \times \mathbb{R}^D \to \mathbb{R}^C$ be a C-class ReLU network with posterior $\mathcal{N}(\boldsymbol{\theta} \mid \boldsymbol{\mu}, \boldsymbol{\Sigma})$ and $\widetilde{f}$ be obtained from $f$ via RGPR over the input space. Suppose that the linearization of $f$ w.r.t. $\boldsymbol{\theta}$ around $\boldsymbol{\mu}$ is employed. For any $\boldsymbol{x}_* \in \mathbb{R}^N$ with $\boldsymbol{x}_* \neq \boldsymbol{0}$ there exists $\beta > 0$ such that for any $\alpha \geq \beta$, the variance of each output component $\widetilde{f}_1(\alpha\boldsymbol{x}_*), \dots, \widetilde{f}_C(\alpha\boldsymbol{x}_*)$ under $p(\widetilde{f}_* \mid \boldsymbol{x}_*, \mathcal{D})$ (8) is in $\Theta(\alpha^3)$.*

As a consequence of Proposition 3, in the binary classification case, the confidence of $\alpha\boldsymbol{x}_*$ decays like $1/\sqrt{\alpha}$ far away from the training data. This can be seen using the (binary) probit approximation. Thus, in this case we obtain the maximum entropy in the limit of $\alpha \to \infty$. In the following theorem we formalize this statement in the more general multi-class classification setting.

**Theorem 4 (Uniform Asymptotic Confidence).** *Let $f : \mathbb{R}^N \times \mathbb{R}^D \to \mathbb{R}^C$ be a C-class ReLU network equipped with the posterior $\mathcal{N}(\boldsymbol{\theta} \mid \boldsymbol{\mu}, \boldsymbol{\Sigma})$ and let $\widetilde{f}$ be obtained from $f$ via RGPR over the*

Table 1: Performances of RGPRs compared to their respective base methods on the detection of far-away outliers. Error bars are standard deviations of ten trials. For each dataset, best values over each vanilla and RGPR-imbued method (e.g. LLL against LLL-RGPR) are in bold.

| Methods | MNIST MMC ↓ | MNIST AUR ↑ | CIFAR10 MMC ↓ | CIFAR10 AUR ↑ | SVHN MMC ↓ | SVHN AUR ↑ | CIFAR100 MMC ↓ | CIFAR100 AUR ↑ |
|---|---|---|---|---|---|---|---|---|
| BNO | 88.2±1.1 | 87.0±2.7 | 22.0±0.2 | 100.0±0.0 | 22.1±0.3 | 100.0±0.0 | 10.5±0.1 | 99.6±0.0 |
| LLL | 99.9±0.0 | 9.8±0.7 | 17.4±0.0 | 100.0±0.0 | 27.5±0.1 | 99.6±0.0 | 5.9±0.0 | 99.9±0.0 |
| LLL-RGPR | **16.6**±0.1 | **100.0**±0.0 | **15.1**±0.1 | 100.0±0.0 | **15.1**±0.0 | **100.0**±0.0 | **4.2**±0.0 | **100.0**±0.0 |
| KFL | 57.2±3.0 | 96.0±0.8 | 69.5±2.5 | 81.2±2.5 | 64.9±2.3 | 90.9±1.6 | 41.1±2.3 | 81.7±1.5 |
| KFL-RGPR | **28.2**±0.2 | **99.8**±0.0 | **27.5**±0.2 | **99.1**±0.1 | **27.5**±0.2 | **99.6**±0.0 | **13.9**±0.1 | **97.5**±0.2 |
| SWAG | 83.2±5.5 | 55.3±20.8 | 50.5±13.1 | 97.4±2.5 | 59.6±9.7 | 97.5±1.9 | 96.0±5.0 | 9.6±11.9 |
| SWAG-RGPR | **27.9**±0.3 | **99.8**±0.1 | **27.6**±0.2 | **99.9**±0.0 | **27.5**±0.1 | **99.9**±0.0 | **17.4**±0.9 | **78.5**±1.6 |
| SVDKL | 59.1±0.6 | 99.7±0.0 | 46.1±0.4 | 99.3±0.1 | 48.5±0.5 | 99.4±0.0 | 55.3±1.8 | 80.9±1.6 |
| SVDKL-RGPR | **22.1**±0.2 | **100.0**±0.0 | **22.1**±0.1 | **100.0**±0.0 | **22.0**±0.1 | **100.0**±0.0 | **9.8**±0.1 | **100.0**±0.0 |

*input space. Suppose that the linearization of $f$ and the generalized probit approximation* (2) *is used for approximating the predictive distribution $p(y_* = c \mid \alpha \boldsymbol{x}_*, \widetilde{f}, \mathcal{D})$ under $\widetilde{f}$. Then for any input $\boldsymbol{x}_* \in \mathbb{R}^N$ with $\boldsymbol{x}_* \neq \boldsymbol{0}$ and for every class $c = 1, \ldots, C$,*

$$\lim_{\alpha \to \infty} p(y_* = c \mid \alpha \boldsymbol{x}_*, \widetilde{f}, \mathcal{D}) = \frac{1}{C}.$$

## 5 RELATED WORK

The mitigation of the asymptotic overconfidence problem has been studied recently. Although Hein et al. (2019) theoretically demonstrated this issue, their proposed method does not fix this issue for $\alpha$ large enough. Kristiadi et al. (2020) showed that any Gaussian-approximated BNN could mitigate this issue even for $\alpha = \infty$. However, the asymptotic confidence estimates of BNNs converge to a constant in $(0, 1)$, not to the ideal uniform confidence. In a non-Bayesian framework, using Gaussian mixture models, Meinke & Hein (2020) integrate density estimates of inliers and outliers data into the confidence estimates of an NN to achieve the uniform confidence far away from the data. Nevertheless, this property has not been previously achieved in the context of BNNs.

Modeling the residual of a predictive model with GP has been proposed by Blight & Ott (1975); Wahba (1978); O'Hagan (1978); Qiu et al. (2020). The key distinguishing factors between RGPR and those methods are (i) RGPR models the residual of BNNs, in contrast to that of point-estimated networks, (ii) RGPR uses a novel kernel which guarantees cubic uncertainty growth, and (iii) RGPR requires no posterior inference. Nevertheless, whenever those methods uses our DSCS kernel, RGPR can be seen as an economical approximation of their posterior: RGPR estimates uncertainty near the data with a BNN, while the GP-DSCS prior estimates uncertainty far away from them.

A combination of weight- and function-space models has been proposed in the context of non-parametric GP posterior sampling. Wilson et al. (2020) proposed to approximate a function as the sum of a weight-space prior and function-space posterior. In contrast, RGPR models a function as the sum of weight-space posterior and function-space prior in the context of parametric BNNs.

## 6 EMPIRICAL EVALUATIONS

Our goal in this section is (i) to validate our analysis in the preceding section: we aim to show that RGPR's low confidence far-away from the training data is observable in practice, and (ii) to explore the effect of the hyperparameters of RGPR to the non-asymptotic confidence estimates. We focus on classification—experiments on regression are in Appendix D.

### 6.1 ASYMPTOTIC REGIME

We use standard benchmark datasets: MNIST, CIFAR10, SVHN, and CIFAR100. We use LeNet and ResNet-18 for MNIST and the rest of the datasets, respectively. Our main reference is the method based on Blight & Ott (1975) (with our kernel): We follow Qiu et al. (2020) for combining the network and GP, and for carrying out the posterior inference. We refer to this baseline as the Blight and Ott method (BNO)—cf. Appendix C for an exposition about this method. The base methods, which RGPR is implemented on, are the following recently-proposed BNNs: (i) last-layer Laplace (LLL, Kristiadi et al., 2020), (ii) Kronecker-factored Laplace (KFL, Ritter et al., 2018), (iii) stochastic weight averaging-Gaussian (SWAG, Maddox et al., 2019), and (iv) stochastic variational deep kernel learning (SVDKL, Wilson et al., 2016). All the kernel hyperparameters for RGPR are set to 1. In all cases, MC-integral with 10 posterior samples is used for making prediction.

To validate Theorem 4, we construct a test dataset artificially by sampling 2000 uniform noises in $[0, 1]^N$ and scale them with a scalar $\alpha = 2000$. The goal is to distinguish test points from these outliers based on the confidence estimates. Since a visual inspection of these confidence estimates as in Figure 1 is not possible in high dimension, we measure the results using the mean maximum confidence (MMC) and area under ROC (AUR) metrics (Hendrycks & Gimpel, 2017). MMC is useful for summarizing confidence estimates, while AUR tells us the usefulness of the confidences for distinguishing between inliers and outliers.

The results are presented in Table 1. We observe that the RGPR-augmented methods are significantly better than their respective base methods. In particular, the confidences drop, as shown by the MMC values. We also observe in Table 3 (Appendix D) that the confidence estimates close to the training data do not significantly change. These two facts together yield high AUR values, close to the ideal value of 100. Moreover, most RGPR-imbued methods achieve similar or better performance to BNO baseline, likely be due to uncertainty already presents in the base BNNs. However, these confidences on far-away points are not quite the uniform confidence due to the number of MC samples used—recall that far away from the data, RGPR yields high variance; since the error of MC-integral depends on both the variance and number of samples, a large amount of samples are needed to get accurate MC-estimates. See Figure 5 (Appendix D) for results with 1000 samples: in this more accurate setting, the convergence to the uniform confidence happens at finite (and small) $\alpha$. Nevertheless, this issue not a detrimental to the detection of far-away outliers, as shown by the AUR values in Table 1.

### 6.2 NON-ASYMPTOTIC REGIME

The main goal of this section is to show that RGPR can also improve uncertainty estimates near the data by varying its kernel hyperparameters. For this purpose, we use a simple hyperparameter optimization using a noise out-of-distribution (OOD) data, similar to Kristiadi et al. (2020), to tune $(\sigma_l^2)$—the details are in Section C.2. We use LLL as the base BNN.

First, we use the rotated-MNIST experiment proposed by Ovadia et al. (2019), where we measure methods' calibration at different rotation angle, see Figure 4. LLL gives significantly better performance than BNO and RGPR improves the performance further. Moreover, we use standard OOD data tasks where one distinguishes in-distribution from out-distribution samples. We do this with CIFAR10 as the in-distribution dataset against various OOD datasets (more results in Appendix D). As shown in Table 2, LLL outperforms for CIFAR10 BNO and RGPR further improves LLL.

## 7 CONCLUSION

We have shown that adding "missing uncertainty" to ReLU BNNs with a carefully-crafted GP prior that represents infinite ReLU features fixes the asymptotic overconfidence problem of such networks. The core of our method is a generalization of the classic cubic-spline kernel, which, when used as the covariance function of the GP prior, yields a marginal variance which scales cubically in the distance between a test point and the training data. Our main strength lies in the simplicity of the proposed method: RGPR is relative straightforward to implement, and can be applied inexpensively to *any* pre-trained BNN. Furthermore, extensive theoretical analyses show that RGPR provides significant improvements to previous results with vanilla BNNs. In particular, we were able to show uniform confidence far-away from the training data in multi-class classifications. On a less formal

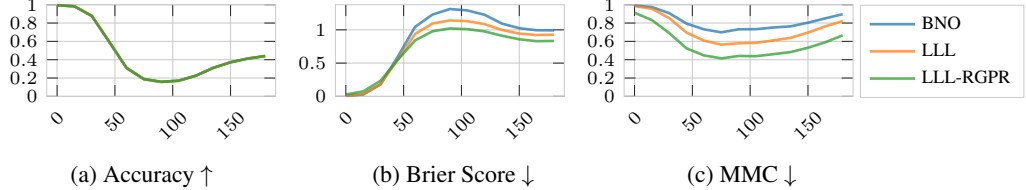

(a) Accuracy ↑          (b) Brier Score ↓          (c) MMC ↓

Figure 4: Rotated-MNIST results (averages of 10 predictions). $x$-axes are rotation angles. In (a), all methods achieve similar accuracies.

Table 2: OOD data detection. Datasets in bold face are the in-distribution datasets.

| Datasets | BNO | | LLL | | LLL-RGPR | |
|---|---|---|---|---|---|---|
| | MMC ↓ | AUR ↑ | MMC ↓ | AUR ↑ | MMC ↓ | AUR ↑ |
| **CIFAR10** | 96.9±0.1 | - | 92.7±0.1 | - | 90.3±0.1 | - |
| SVHN | 69.0±0.0 | 93.6±0.1 | 45.3±0.0 | **96.4**±0.1 | **44.0**±0.0 | 96.0±0.1 |
| LSUN | 76.6±0.0 | 90.8±0.1 | 56.8±0.1 | 92.8±0.1 | **51.5**±0.1 | **93.7**±0.1 |
| CIFAR100 | 80.0±0.0 | 86.3±0.1 | 64.1±0.0 | **88.3**±0.1 | **60.3**±0.0 | **88.3**±0.1 |
| UniformNoise | 75.9±0.4 | 94.3±0.1 | 36.7±0.2 | 99.0±0.0 | **25.4**±0.1 | **99.8**±0.0 |
| Noise | 61.5±0.4 | 96.3±0.1 | 41.8±0.2 | **97.6**±0.1 | **40.5**±0.2 | 97.4±0.1 |

note, our construction, while derived as a *post-hoc* addition to the network, follows a pleasingly simple intuition that bridges the worlds of deep learning and non-parametric/kernel models: The RGPR model amounts to considering a non-parametric model of infinitely many ReLU features, only finitely many of which are trained as a deep ReLU network.

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

## APPENDIX A    DERIVATIONS

### A.1    THE CUBIC SPLINE KERNEL

Recall that we have a linear model $f : [c_{\min}, c_{\max}] \times \mathbb{R}^K \to \mathbb{R}$ with the ReLU feature map $\phi$ defined by $f(x; \boldsymbol{w}) := \boldsymbol{w}^\top \phi(x)$ over the input space $[c_{\min}, c_{\max}] \subset \mathbb{R}$, where $c_{\min} < c_{\max}$. Furthermore, $\phi$ regularly places the $K$ generalized ReLU functions centered at $(c_i)_{i=1}^K$ where $c_i = c_{\min} + \frac{i-1}{K-1}(c_{\max} - c_{\min})$ in the input space and we consider a Gaussian prior $p(\boldsymbol{w}) := \mathcal{N}\left(\boldsymbol{w} \mid \boldsymbol{0}, \sigma^2 K^{-1}(c_{\max} - c_{\min})\boldsymbol{I}\right)$ over the weight $\boldsymbol{w}$. Then, as $K$ goes to infinity, the distribution over the function output $f(x)$ is a Gaussian process with mean 0 and covariance

$$\text{cov}(f(x), f(x')) = \sigma^2 \frac{c_{\max} - c_{\min}}{K} \phi(x)^\top \phi(x') = \sigma^2 \frac{c_{\max} - c_{\min}}{K} \sum_{i=1}^K \text{ReLU}(x; c_i)\text{ReLU}(x'; c_i)$$

$$= \sigma^2 \frac{c_{\max} - c_{\min}}{K} \sum_{i=1}^K H(x - c_i)H(x' - c_i)(x - c_i)(x' - c_i)$$

$$= \sigma^2 \frac{c_{\max} - c_{\min}}{K} \sum_{i=1}^K H(\min(x, x') - c_i)\left(c_i^2 - c_i(x + x') + xx'\right), \qquad (9)$$

where the last equality follows from (i) the fact that both $x$ and $x'$ must be greater than or equal to $c_i$, and (ii) by expanding the quadratic form in the second line.

Let $\bar{x} := \min(x, x')$. Since (9) is a Riemann sum, in the limit of $K \to \infty$, it is expressed by the following integral

$$\lim_{K \to \infty} \text{cov}(f(x), f(x')) = \sigma^2 \int_{c_{\min}}^{c_{\max}} H(\bar{x} - c)\left(c^2 - c(x + x') + xx'\right)\, dc$$

$$= \sigma^2 H(\bar{x} - c_{\min}) \int_{c_{\min}}^{\min\{\bar{x}, c_{\max}\}} c^2 - c(x + x') + xx'\, dc$$

$$= \sigma^2 H(\bar{x} - c_{\min}) \left[\frac{1}{3}(z^3 - c_{\min}^3) - \frac{1}{2}(z^2 - c_{\min}^2)(x + x') + (z - c_{\min})xx'\right]$$

where we have defined $z := \min\{\bar{x}, c_{\max}\}$. The term $H(\bar{x} - c_{\min})$ has been added in the second equality as the previous expression is zero if $\bar{x} \leq c_{\min}$ (since in this region, all the ReLU functions evaluate to zero). Note that

$$H(\bar{x} - c_{\min}) = H(x - c_{\min})H(x' - c_{\min})$$

is itself a positive definite kernel. We also note that $c_{\max}$ can be chosen sufficiently large so that $[-c_{\max}, c_{\max}]^d$ contains for sure the data, e.g. this is anyway true for data from bounded domains like images in $[0, 1]^d$, and thus we can set $z = \bar{x} = \min(x, x')$.

## APPENDIX B    PROOFS

**Proposition 1.** *Suppose $f : \mathbb{R}^N \times \mathbb{R}^D \to \mathbb{R}$ defined by $(\boldsymbol{x}, \boldsymbol{\theta}) \mapsto f(\boldsymbol{x}; \boldsymbol{\theta})$ is a ReLU regression BNN with a prior $p(\boldsymbol{\theta}) = \mathcal{N}(\boldsymbol{\theta} \mid \boldsymbol{0}, \boldsymbol{B})$ and $\mathcal{D} := \{\boldsymbol{x}_m, y_m\}_{m=1}^M$ is a dataset. Let $\widehat{f}^{(0)}$ and $\widetilde{f}$ be defined as in (4), and let $\boldsymbol{x}_* \in \mathbb{R}^N$ be arbitrary. Under the linearization of $f$ w.r.t. $\boldsymbol{\theta}$ around $\boldsymbol{0}$, given that all $\boldsymbol{x}_1, \ldots, \boldsymbol{x}_M$ are sufficiently close to the origin, the GP posterior of $\widetilde{f}_* := \widehat{f}(\boldsymbol{x}_*)$ is given by*

$$p(\widetilde{f}_* \mid \boldsymbol{x}_*, \mathcal{D}) \approx \mathcal{N}(\widetilde{f}_* \mid f(\boldsymbol{x}; \boldsymbol{\mu}), \boldsymbol{g}_*^\top \boldsymbol{\Sigma} \boldsymbol{g}_* + k_\leftrightarrow(\boldsymbol{x}_*, \boldsymbol{x}_*)), \qquad (5)$$

*where $\boldsymbol{\mu}$ and $\boldsymbol{\Sigma}$ are the mean and covariance of the posterior of the linearized network, respectively, and $\boldsymbol{g}_* := \nabla_{\boldsymbol{\theta}} f(\boldsymbol{x}_*; \boldsymbol{\theta})|_{\boldsymbol{0}}$.*

*Proof.* Under the linearization of $f$ w.r.t. $\boldsymbol{\theta}$ around $\boldsymbol{0}$, we have

$$f(\boldsymbol{x}; \boldsymbol{\theta}) \approx \underbrace{f(\boldsymbol{x}; \boldsymbol{0})}_{=0} + \underbrace{\nabla_{\boldsymbol{\theta}} f(\boldsymbol{x}; \boldsymbol{\theta})|_{\boldsymbol{0}}}_{=:\boldsymbol{g}(\boldsymbol{x})}{}^\top \boldsymbol{\theta} = \boldsymbol{g}(\boldsymbol{x})^\top \boldsymbol{\theta}.$$

Now, the definition of RGPR implies that we have

$$\widetilde{f}(\boldsymbol{x}) \approx \boldsymbol{g}(\boldsymbol{x})^\top \boldsymbol{\theta} + \widehat{f}^{(0)}(\boldsymbol{x}); \qquad \widehat{f}^{(0)}(\boldsymbol{x}) \sim \mathcal{N}(0, k_\leftrightarrow(\boldsymbol{x}, \boldsymbol{x})).$$

Following O'Hagan (1978), we thus obtain the following GP prior over $\widetilde{f}$, which marginal is

$$\widetilde{f}(\boldsymbol{x}) \sim \mathcal{N}(\widetilde{f}(\boldsymbol{x}) \mid 0, \boldsymbol{g}(\boldsymbol{x})^\top \boldsymbol{B} \boldsymbol{g}(\boldsymbol{x}) + k_\leftrightarrow(\boldsymbol{x}, \boldsymbol{x})).$$

Suppose we write the dataset as $\mathcal{D} = (\boldsymbol{X}, \boldsymbol{y})$ where $\boldsymbol{X}$ is the data matrix and $\boldsymbol{y}$ is the target vectors, and $\boldsymbol{x}_* \in \mathbb{R}^N$ is an arbitrary test point. Let $\boldsymbol{k}_\leftrightarrow := (k_\leftrightarrow(\boldsymbol{x}_*, \boldsymbol{x}_1), \dots k_\leftrightarrow(\boldsymbol{x}_*, \boldsymbol{x}_M))^\top$, let $\boldsymbol{K}_\leftrightarrow :=$ $(\boldsymbol{K} + \sigma^2 \boldsymbol{I})$ with $K_{ij} := k_\leftrightarrow(\boldsymbol{x}_i, \boldsymbol{x}_j)$ and $\sigma^2 > 0$ sufficiently large be the (regularized) kernel matrix, and let $\boldsymbol{G} := (\boldsymbol{g}(\boldsymbol{x}_1), \dots, \boldsymbol{g}(\boldsymbol{x}_M))$ be the matrix of training "features". As Rasmussen & Williams (2005, Sec. 2.7) suggests, we have then the following GP posterior mean and variance

$$\mathbb{E}(\widetilde{f}(\boldsymbol{x}_*) \mid \mathcal{D}) = \boldsymbol{g}(\boldsymbol{x}_*)^\top \boldsymbol{\mu} + \boldsymbol{k}_\leftrightarrow \boldsymbol{K}_\leftrightarrow^{-1} (\boldsymbol{y} - \boldsymbol{g}(\boldsymbol{x}_*)^\top \boldsymbol{\mu}) \tag{10}$$

$$\mathrm{var}\,(\widetilde{f}(\boldsymbol{x}_*) \mid \mathcal{D}) = k_\leftrightarrow(\boldsymbol{x}_*, \boldsymbol{x}_*) + \boldsymbol{k}_\leftrightarrow^\top \boldsymbol{K}_\leftrightarrow^{-1} \boldsymbol{k}_\leftrightarrow + \boldsymbol{r}^\top (\boldsymbol{B}^{-1} + \boldsymbol{G} \boldsymbol{K}_\leftrightarrow^{-1} \boldsymbol{G}^\top)^{-1} \boldsymbol{r}, \tag{11}$$

where $\boldsymbol{\mu} := (\boldsymbol{B}^{-1} + \boldsymbol{G} \boldsymbol{K}_\leftrightarrow^{-1} \boldsymbol{G}^\top)^{-1} \boldsymbol{G} \boldsymbol{K}_\leftrightarrow^{-1} \boldsymbol{y}$ and $\boldsymbol{r} := \boldsymbol{g}(\boldsymbol{x}_*) - \boldsymbol{G} \boldsymbol{K}_\leftrightarrow^{-1} \boldsymbol{k}_\leftrightarrow$. Since all training points $\boldsymbol{x}_1, \dots, \boldsymbol{x}_M$ are sufficiently close to the origin, by definition of the DSCS kernel, we have $\boldsymbol{k}_\leftrightarrow \approx \boldsymbol{0}$ and $\boldsymbol{K}_\leftrightarrow^{-1} \approx 1/\sigma^2 \boldsymbol{I}$. These imply that

$$\boldsymbol{\mu} \approx (\boldsymbol{B}^{-1} + 1/\sigma^2 \boldsymbol{G} \boldsymbol{G}^\top)^{-1} (1/\sigma^2 \boldsymbol{G} \boldsymbol{y}) \qquad \text{and} \qquad \boldsymbol{r} \approx \boldsymbol{g}(\boldsymbol{x}_*).$$

In particular, notice that $\boldsymbol{\mu}$ is approximately the posterior mean of the Bayesian linear regression on $f$ (Bishop, 2006, Sec. 3.3). Furthermore (10) and (11) become

$$\mathbb{E}(\widetilde{f}(\boldsymbol{x}_*) \mid \mathcal{D}) \approx \boldsymbol{g}(\boldsymbol{x}_*)^\top \boldsymbol{\mu} = f(\boldsymbol{x}_*; \boldsymbol{\mu})$$

$$\mathrm{var}\,(\widetilde{f}(\boldsymbol{x}_*) \mid \mathcal{D}) \approx k_\leftrightarrow(\boldsymbol{x}_*, \boldsymbol{x}_*) + \boldsymbol{g}(\boldsymbol{x}_*)^\top \underbrace{(\boldsymbol{B}^{-1} + 1/\sigma^2 \boldsymbol{G} \boldsymbol{G}^\top)^{-1}}_{=:\boldsymbol{\Sigma}} \boldsymbol{g}(\boldsymbol{x}_*),$$

respectively. Notice in particular that $\boldsymbol{\Sigma}$ is the posterior covariance of the Bayesian linear regression on $f$. Thus, the claim follows. $\qquad \square$

**Proposition 2 (Invariance in Predictions).** *Let $f : \mathbb{R}^N \times \mathbb{R}^D \to \mathbb{R}^C$ be any network with posterior $\mathcal{N}(\boldsymbol{\theta} \mid \boldsymbol{\mu}, \boldsymbol{\Sigma})$ and $\widetilde{f}$ be obtained from $f$ via RGPR (4). Then under the linearization of $f$, for any $\boldsymbol{x}_* \in \mathbb{R}^N$, we have $\mathbb{E}_{p(\widetilde{f}_* \mid \boldsymbol{x}_*, \mathcal{D})} \widetilde{f}_* = \mathbb{E}_{p(f_* \mid \boldsymbol{x}_*, \mathcal{D})} f_*$.*

*Proof.* Simply compare the means of the Gaussians $p(\widetilde{f}_* \mid \boldsymbol{x}_*, \mathcal{D})$ in (8) and $p(f_* \mid \boldsymbol{x}_*, \mathcal{D})$ in (1). $\qquad \square$

To prove Proposition 3 and Theorem 4, we need the following definition. Let $f : \mathbb{R}^N \times \mathbb{R}^D \to \mathbb{R}^C$ defined by $(\boldsymbol{x}, \boldsymbol{\theta}) \mapsto f(\boldsymbol{x}; \boldsymbol{\theta})$ be a feed-forward neural network which use piecewise affine activation functions (such as ReLU and leaky-ReLU) and are linear in the output layer. Such a network is called a ***ReLU network*** and can be written as a continuous piecewise-affine function (Arora et al., 2018). That is, there exists a finite set of polytopes $\{Q_i\}_{i=1}^P$—referred to as ***linear regions*** $f$—such that $\cup_{i=1}^P Q_i = \mathbb{R}^N$ and $f|_{Q_i}$ is an affine function for each $i = 1, \dots, P$ (Hein et al., 2019). The following lemma is central in our proofs below (the proof is in Lemma 3.1 of Hein et al. (2019)).

**Lemma 5 (Hein et al., 2019).** *Let $\{Q_i\}_{i=1}^P$ be the set of linear regions associated to the ReLU network $f : \mathbb{R}^N \times \mathbb{R}^D \to \mathbb{R}^C$, For any $\boldsymbol{x} \in \mathbb{R}^N$ with $\boldsymbol{x} \neq 0$ there exists a positive real number $\beta$ and $j \in \{1, \dots, P\}$ such that $\alpha \boldsymbol{x} \in Q_j$ for all $\alpha \geq \beta$.* $\qquad \square$

**Proposition 3 (Asymptotic Variance Growth).** *Let $f : \mathbb{R}^N \times \mathbb{R}^D \to \mathbb{R}^C$ be a $C$-class ReLU network with posterior $\mathcal{N}(\boldsymbol{\theta} \mid \boldsymbol{\mu}, \boldsymbol{\Sigma})$ and $\widetilde{f}$ be obtained from $f$ via RGPR over the input space. Suppose that the linearization of $f$ w.r.t. $\boldsymbol{\theta}$ around $\boldsymbol{\mu}$ is employed. For any $\boldsymbol{x}_* \in \mathbb{R}^N$ with $\boldsymbol{x}_* \neq \boldsymbol{0}$ there exists $\beta > 0$ such that for any $\alpha \geq \beta$, the variance of each output component $\widetilde{f}_1(\alpha \boldsymbol{x}_*), \dots, \widetilde{f}_C(\alpha \boldsymbol{x}_*)$ under $p(\widetilde{f}_* \mid \boldsymbol{x}_*, \mathcal{D})$ (8) is in $\Theta(\alpha^3)$.*

*Proof.* Let $\boldsymbol{x}_* \in \mathbb{R}^N$ with $\boldsymbol{x}_* \neq \boldsymbol{0}$ be arbitrary. By Lemma 5 and definition of ReLU network, there exists a linear region $R$ and real number $\beta > 0$ such that for any $\alpha \geq \beta$, the restriction of $f$ to $R$ can be written as

$$f|_R(\alpha\boldsymbol{x}; \boldsymbol{\theta}) = \boldsymbol{W}(\alpha\boldsymbol{x}) + \boldsymbol{b},$$

for some matrix $\boldsymbol{W} \in \mathbb{R}^{C \times N}$ and vector $\boldsymbol{b} \in \mathbb{R}^C$, which are functions of the parameter $\boldsymbol{\theta}$, evaluated at $\boldsymbol{\mu}$. In particular, for each $c = 1, \ldots, C$, the $c$-th output component of $f|_R$ can be written by

$$f_c|_R = \boldsymbol{w}_c^\top (\alpha\boldsymbol{x}) + b_c,$$

where $\boldsymbol{w}_c$ and $b_c$ are the $c$-th row of $\boldsymbol{W}$ and $\boldsymbol{b}$, respectively.

Let $c \in \{1, \ldots, C\}$ and let $\boldsymbol{j}_c(\alpha\boldsymbol{x}_*)$ be the $c$-th column of the Jacobian $\boldsymbol{J}(\alpha\boldsymbol{x}_*)$ as defined in (1). Then by definition of $p(\widetilde{f}_* \mid \boldsymbol{x}_*, \mathcal{D})$, the variance of $\widetilde{f}_c|_R(\alpha\boldsymbol{x}_*)$—the $c$-th diagonal entry of the covariance of $p(\widetilde{f}_* \mid \boldsymbol{x}_*, \mathcal{D})$—is given by

$$\mathrm{var}(\widetilde{f}_c|_R(\alpha\boldsymbol{x}_*)) = \boldsymbol{j}_c(\alpha\boldsymbol{x}_*)^\top \boldsymbol{\Sigma} \boldsymbol{j}_c(\alpha\boldsymbol{x}_*) + k_\leftrightarrow(\alpha\boldsymbol{x}_*, \alpha\boldsymbol{x}_*).$$

Now, from the definition of the DSCS kernel in (3), we have

$$k_\leftrightarrow(\alpha\boldsymbol{x}_*, \alpha\boldsymbol{x}_*) = \frac{1}{N} \sum_{i=1}^N k_\leftrightarrow^1(\alpha x_{*i}, \alpha x_{*i})$$

$$= \frac{1}{N} \sum_{i=1}^N \alpha^3 \frac{\sigma^2}{3} x_{*i}^3$$

$$= \frac{\alpha^3}{N} \sum_{i=1}^N k_\leftrightarrow^1(x_{*i}, x_{*i})$$

$$\in \Theta(\alpha^3).$$

Furthermore, we have

$$\boldsymbol{j}_c(\alpha\boldsymbol{x}_*)^\top \boldsymbol{\Sigma} \boldsymbol{j}_c(\alpha\boldsymbol{x}_*) = \left(\alpha(\nabla_{\boldsymbol{\theta}}\boldsymbol{w}_c|_{\boldsymbol{\mu}})^\top \boldsymbol{x} + \nabla_{\boldsymbol{\theta}}b_c|_{\boldsymbol{\mu}}\right)^\top \boldsymbol{\Sigma} \left(\alpha(\nabla_{\boldsymbol{\theta}}\boldsymbol{w}_c|_{\boldsymbol{\mu}})^\top \boldsymbol{x} + \nabla_{\boldsymbol{\theta}}b_c|_{\boldsymbol{\mu}}\right).$$

Thus, $\boldsymbol{j}_c(\alpha\boldsymbol{x}_*)^\top \boldsymbol{\Sigma} \boldsymbol{j}_c(\alpha\boldsymbol{x}_*)$ is a quadratic function of $\alpha$. Therefore, $\mathrm{var}(\widetilde{f}_c|_R(\alpha\boldsymbol{x}_*))$ is in $\Theta(\alpha^3)$. $\quad\square$

**Theorem 4 (Uniform Asymptotic Confidence).** *Let $f : \mathbb{R}^N \times \mathbb{R}^D \to \mathbb{R}^C$ be a $C$-class ReLU network equipped with the posterior $\mathcal{N}(\boldsymbol{\theta} \mid \boldsymbol{\mu}, \boldsymbol{\Sigma})$ and let $\widetilde{f}$ be obtained from $f$ via RGPR over the input space. Suppose that the linearization of $f$ and the generalized probit approximation (2) is used for approximating the predictive distribution $p(y_* = c \mid \alpha\boldsymbol{x}_*, \widetilde{f}, \mathcal{D})$ under $\widetilde{f}$. Then for any input $\boldsymbol{x}_* \in \mathbb{R}^N$ with $\boldsymbol{x}_* \neq \boldsymbol{0}$ and for every class $c = 1, \ldots, C$,*

$$\lim_{\alpha \to \infty} p(y_* = c \mid \alpha\boldsymbol{x}_*, \widetilde{f}, \mathcal{D}) = \frac{1}{C}.$$

*Proof.* Let $\boldsymbol{x}_* \neq \boldsymbol{0} \in \mathbb{R}^N$ be arbitrary. By Lemma 5 and definition of ReLU network, there exists a linear region $R$ and real number $\beta > 0$ such that for any $\alpha \geq \beta$, the restriction of $f$ to $R$ can be written as

$$f|_R(\alpha\boldsymbol{x}) = \boldsymbol{W}(\alpha\boldsymbol{x}) + \boldsymbol{b},$$

where the matrix $\boldsymbol{W} \in \mathbb{R}^{C \times N}$ and vector $\boldsymbol{b} \in \mathbb{R}^C$ are functions of the parameter $\boldsymbol{\theta}$, evaluated at $\boldsymbol{\mu}$. Furthermore, for $i = 1, \ldots, C$ we denote the $i$-th row and the $i$-th component of $\boldsymbol{W}$ and $\boldsymbol{b}$ as $\boldsymbol{w}_i$ and $b_i$, respectively. Under the linearization of $f$, the marginal distribution (8) over the output $\widetilde{f}(\alpha\boldsymbol{x})$ holds. Hence, under the generalized probit approximation, the predictive distribution restricted to $R$ is given by

$$\widetilde{p}(y_* = c \mid \alpha\boldsymbol{x}_*, \mathcal{D}) \approx \frac{\exp(m_c(\alpha\boldsymbol{x}_*)\,\kappa_c(\alpha\boldsymbol{x}_*))}{\sum_{i=1}^C \exp(m_i(\alpha\boldsymbol{x}_*)\,\kappa_i(\alpha\boldsymbol{x}_*))}$$

$$= \frac{1}{1 + \sum_{i \neq c}^C \exp(\underbrace{m_i(\alpha\boldsymbol{x}_*)\,\kappa_i(\alpha\boldsymbol{x}_*) - m_c(\alpha\boldsymbol{x}_*)\,\kappa_c(\alpha\boldsymbol{x}_*)}_{=: z_{ic}(\alpha\boldsymbol{x}_*)})},$$

where for all $i = 1, \ldots, C$,

$$m_i(\alpha \boldsymbol{x}_*) = f_i|_R(\alpha \boldsymbol{x}; \boldsymbol{\mu}) = \boldsymbol{w}_i^\top (\alpha \boldsymbol{x}) + b_i \in \mathbb{R},$$

and

$$\kappa_i(\alpha \boldsymbol{x}) = (1 + \pi/8 \left( v_{ii}(\alpha \boldsymbol{x}_*) + k_\leftrightarrow(\alpha \boldsymbol{x}_*, \alpha \boldsymbol{x}_*) \right))^{-\frac{1}{2}} \in \mathbb{R}_{>0}.$$

In particular, for all $i = 1, \ldots, C$, note that $m(\alpha \boldsymbol{x}_*)_i \in \Theta(\alpha)$ and $\kappa(\alpha \boldsymbol{x})_i \in \Theta(1/\alpha^{\frac{3}{2}})$ since $v_{ii}(\alpha \boldsymbol{x}_*) + k_\leftrightarrow(\alpha \boldsymbol{x}_*, \alpha \boldsymbol{x}_*)$ is in $\Theta(\alpha^3)$ by Proposition 3. Now, notice that for any $c = 1, \ldots, C$ and any $i \in \{1, \ldots, C\} \setminus \{c\}$, we have

$$
\begin{aligned}
z_{ic}(\alpha \boldsymbol{x}_*) &= (m_i(\alpha \boldsymbol{x}_*)\, \kappa_i(\alpha \boldsymbol{x}_*)) - (m_c(\alpha \boldsymbol{x}_*)\, \kappa_c(\alpha \boldsymbol{x}_*)) \\
&= (\underbrace{\kappa_i(\alpha \boldsymbol{x}_*)\, \boldsymbol{w}_i}_{\Theta(1/\alpha^{\frac{3}{2}})} - \underbrace{\kappa_c(\alpha \boldsymbol{x}_*)\, \boldsymbol{w}_c}_{\Theta(1/\alpha^{\frac{3}{2}})})^\top (\alpha \boldsymbol{x}_*) + \underbrace{\kappa_i(\alpha \boldsymbol{x}_*)\, b_i}_{\Theta(1/\alpha^{\frac{3}{2}})} - \underbrace{\kappa_c(\alpha \boldsymbol{x}_*)\, b_c}_{\Theta(1/\alpha^{\frac{3}{2}})}.
\end{aligned}
$$

Thus, it is easy to see that $\lim_{\alpha \to \infty} z_{ic}(\alpha \boldsymbol{x}_*) = 0$. Hence we have

$$
\begin{aligned}
\lim_{\alpha \to \infty} \widetilde{p}(y_* = c \mid \alpha \boldsymbol{x}_*, \mathcal{D}) &= \lim_{\alpha \to \infty} \frac{1}{1 + \sum_{i \neq c}^{C} \exp(z_{ic}(\alpha \boldsymbol{x}_*))} \\
&= \frac{1}{1 + \sum_{i \neq c}^{C} \exp(0)} \\
&= \frac{1}{C},
\end{aligned}
$$

as required. $\qquad \square$

## APPENDIX C  FURTHER DETAILS

### C.1  THE BLIGHT AND OTT'S METHOD

The Blight and Ott's method (BNO) models the residual of polynomial regressions. That is, suppose $\phi : \mathbb{R} \to \mathbb{R}^D$ is a polynomial basis function defined by $\phi(x) := (1, x, x^2, \ldots, x^{D-1})$, $k$ is an arbitrary kernel, and $\boldsymbol{w} \in \mathbb{R}^D$ is a weight vector, BNO assumes

$$\widetilde{f}(x) := \boldsymbol{w}^\top \phi(x) + \widehat{f}(x), \qquad \text{where } \widehat{f}(x) \sim \mathcal{GP}(0, k(x, x)).$$

Recently, this method has been extended to neural network. Qiu et al. (2020) apply the same idea—modeling residuals with GPs—to pre-trained networks, resulting in a method called RIO. Suppose that $f_{\boldsymbol{\mu}} : \mathbb{R}^N \to \mathbb{R}$ is a neural-network with a pre-trained, *point-estimated* parameters $\boldsymbol{\mu}$. Their method is defined by

$$\widetilde{f}(\boldsymbol{x}) := f_{\boldsymbol{\mu}}(\boldsymbol{x}) + \widehat{f}(\boldsymbol{x}), \qquad \text{where } \widehat{f}(\boldsymbol{x}) \sim \mathcal{GP}(0, k_{\text{IO}}(\boldsymbol{x}, \boldsymbol{x})).$$

The kernel $k_{\text{IO}}$ is a sum of RBF kernels applied on the dataset $\mathcal{D}$ (inputs) and the network's predictions over $\mathcal{D}$ (outputs), hence the name IO—input-output. As in the original Blight and Ott's method, RIO also focuses in doing posterior inference on the GP. Suppose that $m(\boldsymbol{x})$ and $v(\boldsymbol{x})$ is the a posteriori marginal mean and variance of the GP, respectively. Then, via standard computations, one can see that even though $f$ is a point-estimated network, $\widetilde{f}$ is a random function, distributed a posteriori by

$$\widetilde{f}(\boldsymbol{x}) \sim \mathcal{N}\left( \widetilde{f}(\boldsymbol{x}) \,\middle|\, \widetilde{f}_{\boldsymbol{\mu}}(\boldsymbol{x}) + m(\boldsymbol{x}), v(\boldsymbol{x}) \right).$$

Thus, BNO and RIO effectively add uncertainty to point-estimated networks.

The posterior inference of BNO and RIO can be computationally intensive, depending on the number of training examples $M$: The cost of exact posterior inference is in $\Theta(M^3)$. While it can be alleviated by approximate inference, such as via inducing point methods and stochastic optimizations, the posterior inference requirement can still be a hindrance for a practical adoption of BNO and RIO, especially on large problems.

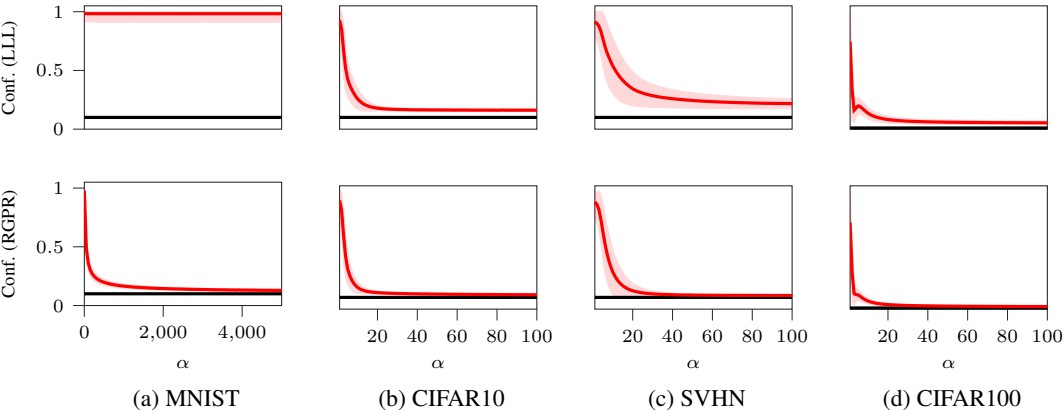

Figure 5: Average confidence as a function of $\alpha$. **Top:** the vanilla LLL. **Bottom:** LLL with RGPR. Test data are constructed by scaling the original test sets with $\alpha$. Error bars are $\pm 1$ standard deviation. Black lines are the uniform confidences. MC-integral with 1000 samples is employed.

## C.2 HYPERPARAMETER TUNING

We have shown in the main text (both theoretically and empirically) that the *asymptotic* performance of RGPR does not depend on the choice of its hyperparameters $(\sigma_l^2)_{l=0}^{L-1}$. Indeed we simply set each $\sigma_l^2$ to its default value 1 for all experiments and showed that RGPR could already fix the asymptotic overconfidence problem effectively.

Nevertheless, Figure 3 gives us a hint that learning these hyperparameters might be beneficial for uncertainty estimation. Intuitively, by increasing $(\sigma_l^2)$, one might be able to make the high confidence (low uncertainty) region more compact. However, if the values of $(\sigma_l^2)$ were too large, the uncertainty will be high even in the data region, resulting in underconfidenct predictions.

Borrowing the contemporary method in robust learning literature (Hendrycks et al., 2019; Hein et al., 2019; Meinke & Hein, 2020, etc.) one way to train $(\sigma_l^2)$ is by using the following min-max objective which intuitively balances high-confidence predictions on inliers and low-confidence predictions on outliers. Let $H$ be the entropy functional, $\mathcal{D}$ the training dataset, $\mathcal{D}_{\text{out}}$ an outlier dataset, $\boldsymbol{\sigma}^2 := (\sigma_l^2)$, and $\lambda \in \mathbb{R}$ be a trade-off parameter. We define:

$$L(\boldsymbol{\sigma}^2) := \mathbb{E}_{\boldsymbol{x}_*^{(\text{in})} \in \mathcal{D}} H\left(\widetilde{p}(y_* \mid \boldsymbol{x}_*^{(\text{in})}, \mathcal{D}; \boldsymbol{\sigma}^2)\right) - \lambda \mathbb{E}_{\boldsymbol{x}_*^{(\text{out})} \in \mathcal{D}_{\text{out}}} H\left(\widetilde{p}(y_* \mid \boldsymbol{x}_*^{(\text{out})}, \mathcal{D}; \boldsymbol{\sigma}^2)\right), \qquad (12)$$

where the predictive distribution $\widetilde{p}(y_* \mid \boldsymbol{x}_*, \mathcal{D}; \boldsymbol{\sigma}^2)$ is as defined in Section 4 with its dependency to $\boldsymbol{\sigma}^2$ explicitly shown. In this paper, for the outlier dataset $\mathcal{D}_{\text{out}}$, we use noise dataset constructed by Gaussian blur and contrast scaling as proposed by Hein et al. (2019). We found that this simple dataset is already sufficient for showing good improvements over the default values ($\sigma_l^2 = 1$). Nevertheless, using more sophisticated outlier datasets, e.g. those used in robust learning literature, could potentially improve the results further. Lastly, we use the trade-off value of $\lambda = 1$ and $\lambda = 0.75$ for our experiments with LeNet/ResNet-18 and DenseNet-BC-121, respectively since we found that $\lambda = 1$ in the latter architecture generally make the network severely underconfident.

## APPENDIX D   ADDITIONAL EXPERIMENTS

### D.1   CLASSIFICATION

We show the behavior of a RGPR-imbued image classifier (LLL) in terms of $\alpha$ in Figure 5. While Table 1 has already shown that RGPR makes confidence estimates close to uniform, here we show that the convergence to low confidence occurred for some small $\alpha$. Furthermore, notice that when $\alpha = 1$, i.e. at the test data, RGPR maintains the high confidence of the base method.

Table 3: Confidence over test sets (i.e. $\alpha = 1$) in term of MMC. Values are averaged over ten trials. Larger is better.

| Methods | MNIST | CIFAR10 | SVHN | CIFAR100 |
|---|---|---|---|---|
| BNO | 99.2±0.0 | 96.9±0.0 | 98.5±0.0 | 82.1±0.1 |
| LLL | 98.5±0.0 | 92.6±0.1 | 91.6±0.0 | 74.6±0.1 |
| LLL-RGPR | 97.9±0.0 | 92.5±0.1 | 91.4±0.0 | 73.8±0.1 |
| KFL | 92.9±0.2 | 86.6±0.1 | 90.8±0.0 | 73.4±0.1 |
| KFL-RGPR | 91.7±0.1 | 86.5±0.1 | 90.7±0.0 | 72.8±0.2 |
| SWAG | 87.6±5.0 | 95.0±0.3 | 97.7±0.2 | 52.9±0.9 |
| SWAG-RGPR | 86.8±4.0 | 94.9±0.2 | 97.6±0.1 | 51.2±1.0 |
| SVDKL | 99.6±0.0 | 97.5±0.0 | 98.6±0.0 | 80.4±0.2 |
| SVDKL-RGPR | 99.6±0.0 | 97.5±0.0 | 98.6±0.0 | 79.9±0.1 |

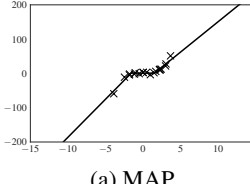
(a) MAP

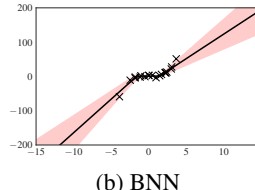
(b) BNN

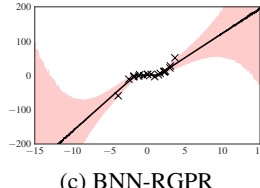
(c) BNN-RGPR

Figure 6: Toy regression with a BNN and additionally, our RGPR. Shades represent ±1 standard deviation.

## D.2 REGRESSION

To empirically validate our method and analysis (esp. Proposition 3), we present a toy regression results in Figure 6. RGPR improves the BNN further: Far-away from the data, the error bar becomes wider.

For more challenging problems, we employ a subset of the standard UCI regression datasets. Our goal here, similar to the classification case, is to compare the uncertainty behavior of RGPR-augmented BNN baselines near the training data (inliers) and far-away from them (outliers). The outlier dataset is constructed by sampling 1000 points from the standard Gaussian and scale them with $\alpha = 2000$. Naturally, the metric we choose is the predictive error bar (standard deviation), i.e. the same metric used in Figure 1. Following the standard practice (see e.g. Sun et al. (2019)), we use a two-layer ReLU network with 50 hidden units. The Bayesian methods used are LLL, KFL, SWAG, and stochastic variational GP (SVGP, Hensman et al., 2015) using 50 inducing points. Finally, we standardize the data and the hyperparameter for RGPR is set to 0.001 so that RGPR does not incur significant uncertainty on the inliers.

The results are presented in Table 4. We can observe that all RGPRs retain high confidence estimates over inlier data and yield much larger error bar compared to the base methods. Furthermore, as we show in Table 5, the RGPR-augmented methods retain the base methods' predictive performances in terms of test RMSE. All in all, these findings confirm the effectiveness of RGPR in far-away outlier detection.

## D.3 NON-ASYMPTOTIC REGIME

Using (12), we show the results of a tuned-RGPR on standard out-of-distribution (OOD) data detection benchmark problems on LeNet/ResNet architecture in Tables 2 and 6. Furthermore, we show results for deeper network (121-layer DenseNet-BC) in Table 7. We optimize $(\sigma_l^2)$ using Adam with learning rate 0.1 over each validation set and the noise dataset (both contain 2000 points) for 10 epochs. Note that this process is quick since no backpropagation over the networks is required. In general tuning the kernel hyperparameters of RGPR lead to significantly lower average confidence (MMC) over outliers compared to the vanilla method (LLL) which leads to higher detection

Table 4: Regression far-away outlier detection. Values correspond to predictive error bars (averaged over ten trials), similar to what shades represent in Figures 1 and 2. "In" and "Out" correspond to inliers and outliers, respectively.

| Methods | housing In ↓ | housing Out ↑ | concrete In ↓ | concrete Out ↑ | energy In ↓ | energy Out ↑ | wine In ↓ | wine Out ↑ |
|---|---|---|---|---|---|---|---|---|
| LLL | 0.405 | 823.215 | 0.324 | 580.616 | 0.252 | 319.890 | 0.126 | 24.176 |
| LLL-RGPR | 0.407 | **2504.325** | 0.329 | **3394.466** | 0.253 | **2138.909** | 0.129 | **1948.813** |
| KFL | 1.171 | 2996.606 | 1.281 | 2518.338 | 0.651 | 1486.748 | 0.291 | 475.141 |
| KFL-RGPR | 1.165 | **3909.140** | 1.264 | **4258.177** | 0.656 | **2681.780** | 0.292 | **2031.481** |
| SWAG | 0.181 | 440.085 | 1.192 | 2770.455 | 0.418 | 1066.044 | 0.181 | 77.357 |
| SWAG-RGPR | 0.186 | **2403.366** | 1.146 | **4693.273** | 0.428 | **2647.922** | 0.187 | **1947.677** |
| SVGP | 0.641 | 2.547 | 0.845 | 3.100 | 0.367 | 2.237 | 0.092 | 0.983 |
| SVGP-RGPR | 0.641 | **1973.506** | 0.845 | **1932.061** | 0.367 | **1931.299** | 0.095 | **1956.027** |

Table 5: The corresponding predictive performance to Table 4 in terms of the RMSE metric. Values are averaged over ten trials. Smaller is better.

| Methods | housing | concrete | energy | wine |
|---|---|---|---|---|
| LLL | **7.361**±3.050 | **42.039**±6.260 | **6.228**±1.864 | **0.423**±0.048 |
| LLL-RGPR | **7.361**±3.050 | **42.039**±6.260 | **6.228**±1.864 | **0.423**±0.048 |
| KFL | **8.549**±2.685 | **42.729**±6.355 | **6.427**±1.912 | **0.421**±0.052 |
| KFL-RGPR | **8.466**±2.732 | **42.608**±6.400 | **6.413**±1.908 | **0.422**±0.050 |
| SWAG | **7.265**±3.008 | **39.308**±4.481 | **2.481**±0.755 | **0.429**±0.047 |
| SWAG-RGPR | **7.274**±3.016 | **39.459**±4.626 | **2.469**±0.760 | **0.430**±0.047 |
| SVGP | **14.512**±4.751 | **50.657**±7.193 | **5.697**±1.443 | **0.381**±0.057 |
| SVGP-RGPR | **14.512**±4.751 | **50.657**±7.193 | **5.697**±1.443 | **0.381**±0.057 |

performance (AUR). Finally, we show the calibration performance of RGPR on the DenseNet in Table 8. We observe that the base BNN we use, LLL, does not necessarily give good calibration performance. Applying RGPR improves this, making LLL better calibrated than the "gold standard" baseline BNO.

We also compare LLL-RGPR to Deep Ensemble (DE) (Lakshminarayanan et al., 2017) which has been shown to perform better compared to Bayesian methods (Ovadia et al., 2019). As we can see in Table 10, LLL-RGPR is competitive to DE. These results further reinforce our finding that RGPR is also useful in non-asymptotic regime.

Inspecting the optimal hyperparameters ($\sigma_l^2$), we found that high kernel variances on higher layers tend to be detrimental to the uncertainty estimate, as measured by (12), leading to low variance values on those layers, cf. Table 9. Specifically, for the LeNet architecture, we found that having high kernel variance on the input (the bottom-most layer) is desirable. Meanwhile, the first residual block and the second dense block are the most impactful in terms of uncertainty estimation for the ResNet and DenseNet architectures, respectively.

Table 6: OOD data detection results using the hyperparameter tuning objective in (12). All values are averages and standard deviations over 10 trials.

| | BNO | | LLL | | LLL-RGPR | |
|---|---|---|---|---|---|---|
| Datasets | MMC ↓ | AUR ↑ | MMC ↓ | AUR ↑ | MMC ↓ | AUR ↑ |
| **MNIST** | 99.2±0.0 | - | 98.5±0.0 | - | 91.2±0.1 | - |
| EMNIST | 82.3±0.0 | 89.2±0.1 | 70.3±0.0 | **92.1**±0.1 | **54.9**±0.0 | 91.7±0.1 |
| FMNIST | 66.3±0.0 | 97.4±0.0 | 56.1±0.0 | 98.3±0.0 | **37.0**±0.0 | **98.9**±0.0 |
| GrayCIFAR10 | 48.0±0.0 | 99.7±0.0 | 41.2±0.0 | 99.7±0.0 | **24.0**±0.0 | **99.9**±0.0 |
| UniformNoise | 96.7±0.0 | 95.2±0.1 | 92.3±0.1 | 94.6±0.1 | **43.3**±0.2 | **99.2**±0.0 |
| Noise | **12.9**±0.1 | **100.0**±0.0 | **12.9**±0.1 | **100.0**±0.0 | 12.8±0.1 | **100.0**±0.0 |
| **SVHN** | 98.5±0.0 | - | 91.6±0.0 | - | 88.8±0.0 | - |
| CIFAR10 | 70.9±0.0 | 95.0±0.0 | 40.7±0.0 | 97.2±0.0 | **36.5**±0.0 | **97.7**±0.0 |
| LSUN | 72.2±0.0 | 95.1±0.0 | 41.5±0.2 | 97.3±0.0 | **36.2**±0.1 | **98.0**±0.1 |
| CIFAR100 | 71.8±0.0 | 94.1±0.0 | 42.1±0.0 | 96.7±0.0 | **37.7**±0.0 | **97.3**±0.0 |
| UniformNoise | 68.9±0.7 | 96.6±0.2 | 40.0±0.5 | 97.6±0.1 | **31.7**±0.3 | **98.8**±0.0 |
| Noise | 66.5±0.5 | 96.3±0.2 | 36.3±0.3 | **97.9**±0.1 | **34.8**±0.3 | 97.8±0.1 |
| **CIFAR100** | 82.2±0.2 | - | 74.6±0.2 | - | 69.5±0.2 | - |
| SVHN | 46.8±0.0 | **84.4**±0.2 | 42.7±0.0 | 80.3±0.2 | **40.0**±0.0 | 78.1±0.2 |
| LSUN | 53.5±0.0 | 80.3±0.2 | 39.8±0.1 | 82.6±0.2 | **33.0**±0.1 | **83.7**±0.2 |
| CIFAR10 | 56.0±0.0 | 78.1±0.2 | 44.4±0.0 | **78.9**±0.2 | **38.8**±0.0 | **79.0**±0.2 |
| UniformNoise | 31.6±0.4 | 93.3±0.2 | 21.4±0.2 | 94.7±0.2 | **9.6**±0.1 | **99.1**±0.1 |
| Noise | 51.9±0.6 | **81.3**±0.4 | 47.5±0.7 | 77.1±0.5 | 44.4±0.6 | 74.8±0.5 |

Table 7: OOD data detection results using the hyperparameter tuning objective in (12) on DenseNet-BC-121 network. All values are averages and standard deviations over 10 trials.

| | BNO | | LLL | | LLL-RGPR | |
|---|---|---|---|---|---|---|
| Datasets | MMC ↓ | AUR ↑ | MMC ↓ | AUR ↑ | MMC ↓ | AUR ↑ |
| **CIFAR10** | 96.5±0.1 | - | 98.1±0.1 | - | 91.1±0.1 | - |
| SVHN | 86.7±0.0 | 83.5±0.1 | 83.2±0.0 | **91.3**±0.1 | **68.8**±0.0 | 83.6±0.2 |
| LSUN | 78.3±0.1 | 89.3±0.1 | 80.6±0.1 | 91.3±0.1 | **48.3**±0.3 | **94.5**±0.2 |
| CIFAR100 | 82.4±0.0 | 86.0±0.1 | 85.2±0.0 | 88.9±0.1 | **57.5**±0.0 | **90.3**±0.2 |
| UniformNoise | 90.6±0.3 | 83.9±0.3 | 80.2±0.5 | 94.2±0.1 | **51.4**±0.4 | **94.9**±0.1 |
| Noise | 75.7±0.6 | 92.8±0.1 | 72.1±0.4 | **96.5**±0.1 | 57.2±0.3 | 92.8±0.1 |
| **SVHN** | 89.9±0.0 | - | 97.8±0.0 | - | 93.9±0.0 | - |
| CIFAR10 | **47.5**±0.0 | 90.3±0.0 | 69.5±0.0 | 94.1±0.0 | 48.9±0.0 | **96.4**±0.0 |
| LSUN | **38.7**±0.1 | 92.9±0.1 | 67.7±0.0 | 94.7±0.0 | 46.1±0.2 | **97.3**±0.1 |
| CIFAR100 | **48.1**±0.0 | 89.9±0.0 | 69.1±0.0 | 93.8±0.0 | 48.5±0.0 | **96.3**±0.0 |
| UniformNoise | 65.9±0.5 | 85.8±0.1 | 63.6±0.5 | **97.3**±0.1 | **50.4**±0.4 | 97.2±0.1 |
| Noise | 32.0±0.3 | 95.2±0.1 | 30.7±0.3 | **99.5**±0.0 | **29.2**±0.3 | 99.0±0.1 |
| **CIFAR100** | 85.1±0.1 | - | 89.3±0.1 | - | 71.0±0.1 | - |
| SVHN | 70.8±0.0 | 72.8±0.2 | 74.0±0.0 | **76.1**±0.1 | **56.4**±0.0 | 67.0±0.1 |
| LSUN | 67.5±0.1 | 75.9±0.2 | 69.5±0.1 | 79.9±0.2 | **37.4**±0.2 | **84.5**±0.2 |
| CIFAR10 | 72.3±0.0 | 70.6±0.1 | 74.8±0.0 | **74.9**±0.1 | **48.3**±0.0 | 74.8±0.1 |
| UniformNoise | 78.8±0.3 | 71.0±0.3 | 71.2±0.3 | **82.4**±0.2 | **42.4**±0.2 | 81.6±0.2 |
| Noise | 87.3±0.2 | 57.2±0.4 | 90.0±0.2 | **60.5**±0.5 | 76.4±0.4 | 45.2±0.5 |

Table 8: Calibration performance of RGPR on DenseNet-BC-121. Values are expected calibration errors (ECEs), averaged over ten prediction runs. RGPR makes the base BNN (LLL) more calibrated—even more than the "gold standard GP" in BNO.

| Datasets | BNO | LLL | LLL-RGPR |
|---|---|---|---|
| CIFAR10 | 17.053±0.637 | 20.758±0.745 | **7.451**±**0.200** |
| SVHN | 6.281±0.092 | 5.705±0.174 | **4.929**±**0.110** |
| CIFAR100 | 22.423±0.416 | 23.630±0.430 | **6.244**±**0.216** |

Table 9: Optimal hyperparameter for each layer (or residual block and dense block for ResNet and DenseNet, respectively) on LLL.

| Datasets | Input | Layer 1 | Layer 2 | Layer 3 | Layer 4 |
|---|---|---|---|---|---|
| **LeNet & ResNet-18** | | | | | |
| MNIST | 2.596e+02 | 1.489e+00 | 2.563e-02 | 2.219e-02 | - |
| CIFAR10 | 9.093e-03 | 3.173e+02 | 2.178e-01 | 1.229e-02 | 7.894e-03 |
| SVHN | 9.009e-03 | 5.559e+02 | 5.507e+00 | 1.390e-02 | 9.039e-03 |
| CIFAR100 | 7.897e-03 | 1.001e+02 | 4.254e-01 | 9.106e-02 | 7.455e-03 |
| **DenseNet-BC-121** | | | | | |
| CIFAR10 | 1.635e-02 | 1.025e-01 | 1.102e+03 | 8.886e+00 | 2.476e+00 |
| SVHN | 7.224e-03 | 3.798e-02 | 1.113e+02 | 2.909e+00 | 7.807e-03 |
| CIFAR100 | 1.016e-02 | 5.203e-02 | 1.233e+02 | 3.335e+00 | 6.716e-02 |

Table 10: Comparison between Deep Ensemble (DE) and LLL-RGPR in terms of AUR. Results for DE are obtained from (Meinke & Hein, 2020) since we use the same networks and training protocol.

| Datasets | DE | LLL-RGPR |
|---|---|---|
| **MNIST** | | |
| EMNIST | 92.1 | 91.7±0.1 |
| FMNIST | **99.2** | 98.9±0.0 |
| GrayCIFAR10 | **100.0** | 99.9±0.0 |
| UniformNoise | 97.9 | **99.2**±0.0 |
| Noise | **100.0** | **100.0**±0.0 |
| **CIFAR10** | | |
| SVHN | 90.3 | **96.0**±**0.1** |
| LSUN | 92.0 | **93.7**±0.1 |
| CIFAR100 | 88.2 | **88.3**±0.1 |
| UniformNoise | 96.6 | **99.8**±0.0 |
| Noise | 90.3 | **97.4**±**0.1** |
| **SVHN** | | |
| CIFAR10 | **97.9** | 97.7±0.0 |
| LSUN | 97.9 | **98.0**±0.1 |
| CIFAR100 | **97.6** | 97.3±0.0 |
| UniformNoise | 95.6 | **98.8**±0.0 |
| Noise | **98.2** | 97.8±0.1 |
| **CIFAR100** | | |
| SVHN | **83.2** | 78.1±0.2 |
| LSUN | 81.6 | **83.7**±0.2 |
| CIFAR10 | 76.3 | **79.0**±0.2 |
| UniformNoise | 36.6 | **99.1**±0.1 |
| Noise | 67.5 | **74.8**±**0.5** |

