# OpenReview forum: "Fixing Asymptotic Uncertainty of Bayesian Neural Networks with Infinite ReLU Features"
_ICLR.cc/2021/Conference — Reject_

### Official Review · AnonReviewer2 · 2020-10-28
**Interesting idea for covariance correction to improve UQ**

**Rating:** 5
**Confidence:** 4

**Review:**

Working under the Bayesian Neural Network setting, the authors proposed a way to inflate the resulting posterior covariance, so as to get improved predictive uncertainty quantification for new data points that are far from the training set, while also maintain similar performance when these new data are close to the training set.

Overall I think the writing is clear, but I had to revisit previous sections in order to tie-up and understand all the various approximation schemes used. I find the derivation of the double-sided cubic spline and the use of infinitely many ReLU's to boost the posterior covariance novel and interesting. This method has potential to be applied to other learning algorithms and get improved confidence statement, such as Variational Bayes where it is known to produce overconfident output.

Despite the authors' claim of doing an extensive theoretical analysis, I find that theoretical arguments quite heuristic in some places. It would be more informative to quantify the approximation errors incurred when using the various approximation methods discussed in this paper, in particular, network linearlization through Taylor's theorem. It seems to me that you just treating the neural network $f$ as a typical differentiable function and ignore the network structure within it by just doing $\approx$.

The level-wise RGPR covariance kernel in (6) does not seem to be correct. For level 1, you have
$\boldsymbol{h}^{(1)} (\boldsymbol{x}_{*})$.

For level 2 however, $\boldsymbol{h}^{(2)}$ is obtained by $g(W\boldsymbol{h}^{(1)}+\boldsymbol{c})$ where $W$ is the weight matrix at level 1, $\boldsymbol{c}$ is the bias and $g$ is some activation function applied entry-wise, e.g., ReLU. Since the entries of $W$ is part of $\boldsymbol{\theta}$ the parameter for the entire network, $W$ is random because $\boldsymbol{\theta}$ is assigned a prior. This implies that $\boldsymbol{h}^{(1)}$ and $\boldsymbol{h}^{(2)}$ are dependent and likewise for higher levels. Hence the covariance kernel of $\hat{f}=\hat{f}^{(0)}+\cdots+\hat{f}^{(L-1)}$ is not the just them sum of the individual kernels but something more complicated because they are now dependent due to $\boldsymbol{h}_{*}$. Can the authors please clarify this?

I find the addition of a kernel function to do correction quite ad-hoc and not very Bayesian. Is it possible to incorporate this term into the prior?

Some other comments:
1. Section 2.2, line 6, $c_d$ should be $c_D$. Also, is this the same $D$ for the dimension of $\boldsymbol{\theta}$ the network weights? Then taking $D\to\infty$ means you have infinite weights?

2. (3) does not seem to cover $0$

3. In Proposition 1, $\boldsymbol{\mu}$ and $\mathrm{\Sigma}$ are the mean and covariance of the approximate posterior of $\boldsymbol{\theta}$, and for $\boldsymbol{g}_{*}$, $\boldsymbol{0}$ should be $\boldsymbol{\mu}$

---

> ### Author Response · Authors · 2020-11-15
> **Response to AnonReviewer2**
>
> We appreciate the feedback---we have incorporated all the suggestions into the paper.
>
> First of all, we would like to re-emphasize that RGPR is not just “inflating” the predictive variance in an ad-hoc manner. RGPR is deeply motivated by the connection between GPs and ReLU BNNs. We refer the reviewer to our response to AnonReviewer4, point 3.
>
>
> (1.) “The approximation errors are not shown; treating the neural network  as a typical differentiable function and ignoring the network structure within it?”
>
> We deliberately chose to not discuss the approximations extensively since they are standard and have been extensively discussed in literature. For example, discussion regarding network linearization and (binary) probit approximation can be found in standard textbooks like Bishop’s. The generalized probit approximation is discussed in Gibbs’ work [1] and recently in [2]. Their errors have also been empirically quantified before. Please refer to Fig. 1 of [3] for the probit approximation. Furthermore, it has recently also been argued that network linearization is the correct way of making predictions in some cases [5].
>
> Please note that the network linearization is done in the parameter space: We view the network as a function of its parameter given that we have a fixed input $x$. The linear approximation therefore does not affect the network structure since it does not affect the network as a function from $x$ to $f(x)$. In other words, all layer-wise structure and non-linearities are preserved. This can be seen clearly in Eq. 1: The mean of the distribution is obtained by doing the standard forward pass of $f$ on $x$, using $\theta = \mu$.
>
>
> (2.) “The level-wise RGPR covariance kernel in (6) does not seem to be correct; $h^{(1)}, \dots, h^{(L)}$ are independent?”
>
> Thank you for noticing this, we agree that this can be made clearer---we have added more explanation in the text (see also Alg. 1). Technically, any learned representation of the input can be used. The key reason why we consider representations of the input is to adapt our kernel to the data region (Fig. 3), and nothing more. In practice, it is convenient to use the point-estimate version of the network to obtain this representation. Concretely, given a posterior $\mathcal{N}(\theta \mid \mu, \Sigma)$, we use the fixed network $f(x; \mu)$ to obtain $h^{(1)}, \dots, h^{(L)}$. These representations are thus not random. In any case, the independence assumption on $h^{(1)}, \dots, h^{(L)}$ does not pose any problem since it is expressed in the prior.
>
>
> (3.) “I find the addition of a kernel function to do correction quite ad-hoc and not very Bayesian. Is it possible to incorporate this term into the prior?”
>
> Eq. 4 follows from the standard way of modeling predictive residuals in Bayesian literature [4, Sec. 2.7]. Note that, while the final RGPR algorithm can be implemented on any pre-trained BNNs, in its derivation (Sec. 3.2) we do not add a GP _post-hoc_, but include it in the prior, jointly with a prior over the weight-space of the neural network, and then show that it is in fact possible to do approximately exact full Bayesian inference in this nonparametric model at nearly no cost overhead (Prop. 1). We would argue that this is a very natural Bayesian treatment indeed. But perhaps we misunderstand. Could you perhaps clarify during the rebuttal in which sense it is "not very Bayesian"?
>
>
> (4.) Additional questions.
> * “Section 2.2, line 6?”: Indeed you are right, there is a typo and $D$ there is unclear. Here $D$ is the number of ReLU features of a linear classifier $g$. As $D \to \infty$, the weight vector $w$ also becomes infinitely long. We have made this clearer in the text.
> * “(3) does not seem to cover $0$?”: $k_\leftrightarrow$ does cover the origin. Its value there is zero.
>
>
> Refs.
> 1. Bishop. Pattern recognition and machine learning. Springer, 2006.
> 2. Gibbs. Bayesian Gaussian processes for regression and classification.
> 3. Lu et al. "Uncertainty Estimation with Infinitesimal Jackknife, Its Distribution and Mean-Field Approximation."
> 4. Rasmussen. "Gaussian processes in machine learning."
> 5. Immer et al. "Improving predictions of Bayesian neural networks via local linearization."

---

### Official Review · AnonReviewer4 · 2020-10-28
**Initial review - updated**

**Rating:** 5
**Confidence:** 4

**Review:**

**SUMMARY**
The authors consider the issue of overconfidence in ReLU NN and BNNs, particularly for data that are far (in Euclidean distance) from the training data. They address this by modeling the residual (to the NN) in the latent space with a GP. The kernel for this GP is derived as the limit of infinitely many ReLU-based random features. Specifically, this kernel has the property that it scales cubically with the norm of the input, and so causes large uncertainty away from the origin. Crucially, the GP term changes little from its prior distribution after conditioning on the data, so no expensive inference is required under the approximation made.

**PROS**
The authors’ proposal seems to improve the overconfidence of existing methods on the tasks they study. While there is a lot of technical motivation, the paper is easy to follow and results in a simple and efficient modification to existing methods.

**CONS**
The main weak point of the paper is the experimental section. The OOD detection task of distinguishing a test set from uniform noise, while perhaps an interesting proof of principle, does not seem very relevant in practice. The results from Table 2 are mixed when compared to the baseline LLL in tasks with real OOD data (as opposed to noise).

Some of the theoretical results and proofs are quite informal or vague. For example in Proposition 1: “are close enough to the origin”

**RECOMMENDATION**
In its current form, I vote for rejecting the paper. First, while the theoretical motivation for the method is nice, many approximations are used to produce such a simple form and more analysis is needed to establish what in particular is advantageous about the DSCS kernel over other kernels. Second, the experimental results are quite restricted and should be expanded. While seeing what the method adds to some baselines is very reasonable, it would be useful to have some discussion comparing results to the current SotA.

**ADDITIONAL QUESTIONS**
The significance of the kernel derived from the limit of ReLU features is unclear to me. The key properties seem to be that the kernel is small when one argument is near the origin and that it is \omega(\alpha^2). Would another kernel with these properties work just as well?

What is the baseline performance (in terms of accuracy or likelihood) for the methods considered in Table 1?

**MINOR COMMENTS**
As there may be images that are very far apart but are nevertheless quite similar, it’s unclear why Euclidean distance in the input space is important. More motivation for this would be helpful.

- In eq (1), f_\mu is not defined. Should this be just \mu?
- In section 2.2 both d and D are used.
- Section 3.1: “which is one-zero only on (0, \inf)” Which argument does this refer to?
- “the value of k(x*,x*) is cubic in x*”: What does this mean? x* is a vector.
- The notation f ~ GP(f | mu, V) is unnecessary. f ~ GP(mu, V) is sufficient.
- “Under this assumption, give an input x* it is clear…”: It does affect the output of the BNN due to the link function.
- Figure 3d is reference, but does not exist.

---

> ### Author Response · Authors · 2020-11-15
> **Response to AnonReviewer4**
>
> Thank you for your feedback. We will address your concerns and questions below. We have incorporated your suggestions and made revision to the manuscript.
>
>
> (1.) “The experiment in Table 1 is irrelevant in practice?”
>
> This experiment is useful to show that RGPR can detect extreme failure events---cases where vanilla ReLU networks and even standard Bayesian ReLU networks could fail [1, 2]. While we agree that they might not frequently occur in practice, in safety-critical systems we need reliability in all cases. In Table 4 in the appendix we additionally show “real-world cases” via UCI benchmark datasets where the networks are particularly vulnerable since their input spaces, unlike image space, are unbounded.
>
>
> (2.) “The results in Table 2 is mixed; Comparison with the current SoTA?”
>
> We refer the reviewer to the full results in Table 6 (appendix). There, we show that RGPR consistently improves LLL across all datasets. We did not put these results in the main text due to space limitation. But we will gladly do so if necessary.  Moreover, while attaining SotA in OOD detection is besides the point of our paper, we have added comparison with Deep Ensemble which have been previously shown to perform the best among Bayesian methods in terms of uncertainty quantification [3]. The results are in Table 10 (appendix): LLL-RGPR is competitive for OOD to DE, while being computationally cheaper.
>
>
> (3.) “Relies on many approximations, why the DSCS kernel?”
>
> We rely on approximations, such as network linearization and the generalized probit approximation, for our analysis since otherwise it would be intractable.
>
> Here, we would like to re-emphasize how the DSCS kernel arises and why it is natural to use it over other kernels. We can think of GP as regularly placing an infinite number features (e.g. a Gaussian feature function, for the squared-exponential kernel) and apply a prior weight on each of these features. In the case of GP with ReLU features, one can show (discussed in Sec. 2.2) that we arrive at the cubic-spline kernel. In the meantime, parametric ReLU BNNs can be seen as only using a fraction of these features---those that “cover” the data. This is why these BNNs have residuals in their uncertainty and thus can still be overconfident away from the data since they have no means for explaining what happens outside the data region. So, the natural thing to do is to also consider the infinitely many features that are not used by BNNs. This can be done by modeling the residual of BNNs as shown in Eq. 4. Nevertheless, the 1D cubic-spline kernel is only non-zero for $x > 0$---intuitively, it only places ReLU features on the positive subset of the real line, with their linear terms oriented to the right (Fig. 2)---thus it does not cover the whole space. The DSCS kernel is thus proposed to avoid this issue. This construction is particularly advantageous since it has negligible values close to the origin which allows us to avoid costly GP posterior inference (illustrated by Prop. 1), and it has a super-quadratic variance growth away from the origin which allows us to obtain uniform confidence away from the data (Sec. 4). Lastly, while there might exist other kernels that can achieve similar properties, the DSCS kernel is a natural kernel to use for ReLU BNNs since it is based on the cubic-spline kernel.
>
>
> (4.) “Prop 1: what does close enough to the origin mean?”
>
> “Close enough to the origin” in this context means that by sufficiently shifting and scaling the dataset, we can guarantee that for any $\epsilon > 0$, for all $x$ in the dataset, the kernel value $k_\leftrightarrow(x, x)$ is at most some $\epsilon$.
>
>
> (5.) “Baseline performance for the methods in Table 1?”
>
> All the methods achieve similar performances, around: 99%, 92%, 96%, and 60% for MNIST, CIFAR10, SVHN, and CIFAR100, respectively.
>
>
> (6.) “It’s unclear why Euclidean distance in the input space is important?”
>
> It is not important per se. Other distance measures would be possible, we could carry our analyses out w.r.t. any distance metric induced by an $\ell_p$-norm.
>
>
> (7.) Minor comments
>
> - “$\mu$ in Eq. 1”: $f_\theta$ is defined in the first line of Sec. 2.1. Note that here $\theta$ is a random variable. $f_\mu$ means the particular $f$ where $\theta = \mu$.
> - “Section 3.1, $(0, \infty)$”: It refers to $k_\rightarrow^1$.
> - “The value of $k_\leftrightarrow(x_*,x_*)$ is cubic in $x_*$”: We meant that $k_\leftrightarrow$ is cubic in the sense that for any $\alpha \in \mathbb{R}$, we have $k_\leftrightarrow(\alpha x_*, \alpha x_*) \in \Theta(\alpha^3)$.
>
>
> Refs.
> 1. Hein et al. "Why ReLU networks yield high-confidence predictions far away from the training data and how to mitigate the problem."
> 2. Kristiadi et al. "Being Bayesian, Even Just a Bit, Fixes Overconfidence in ReLU Networks."
> 3. Ovadia, et al. "Can you trust your model's uncertainty? Evaluating predictive uncertainty under dataset shift."

---

### Official Review · AnonReviewer1 · 2020-10-30
**Interesting but may be overcomplicated**

**Rating:** 7
**Confidence:** 3

**Review:**

Update during rebuttal: I have read other reviews and authors' answers to them. My main concern about the approach to be potentially overcomplicated has been address and I was convinced. I am raising the score for the paper.
/============================================================================================================

The paper addresses the issue of overconfidence of neural network outputs. Pointing to the recent results, the authors state that even Bayesian neural networks (BNNs) are overconfident in their predictions far away from the training data. They propose a method how to fix that. The method is de-facto a simple post-hoc procedure but it is backed up with theory and theoretical guarantees.

Stong points:
* Interesting idea of extending BNNs with GP to address uncertainty in the areas far away from training data
* The idea is derived theoretically and theoretical guarantees are proved
* The method is shown to work empirically in rather extensive experiments

Weak points:
* My biggest and basically only concern about the proposed method is whether it is too complicated. Why would one need to derive GP via these infinite ReLU feature maps? What stops just to declare the usage of cubic spline kernel? This indeed allows the authors to beautifully connect finite BNN with infinite GP with the statement like the last one in the paper, but is there other motivation for that?
Moreover, if the proposed addition to a pre-trained BNN does not depend on training data, how would it compare to something very rough and straightforward, like just adding to variance of the BNN a function that would cubically grow with the distance away from training data?
In any case, I believe it would be interesting to add this kind of ablation study to empirical comparison.

I am voting to weak acceptance but willing to upgrade the score if the authors provide convincing evidence about necessity of all components of the proposed method.

I am voting to accept because despite of the mentioned above concerns of the method to be overcomplicated, the overall idea is interesting, the paper is well-written and easy to follow, the paper addresses the important problem that has impact for the wide audience, the proposed method provides theoretical guarantees while in practice being used as a simple post-hoc procedure.

Questions to authors:
Could you please clarify why you need deriving GP via infinite ReLU maps and why the proposed method would be better than the rough fix to the problem of overconfidence far away from the training data described above?

Additional comments/suggestions (not too important for evaluation but might be used to improve the paper):
1.	ReLU is not defined
2.	ReLU features are only introduced in Section 2.2 although used in Introduction
3.	First paragraph in Section 2.1. For the full picture, some of the definitions are missing, for example, x, y, N, D, and C
4.	Maybe to avoid using expression: “Bayesian methods, which turn standard networks into Bayesian neural networks”
5.	Section 3.1. What about zeros? Two kernels covers (-\infty, 0) and (0, \infty) where 0 is excluded in both
6.	DSCS abbreviation in caption to figure 2 appears before its definition as reference to figure 2 is before this definition
7.	Figure 3 is unclear. The authors should probably elaborate the caption.
8.	What is n in second paragraph in Section 6.1?
9.	Figure 4.a – does this mean that all 3 lines coincide? Or where the other 2 lines? If the former, at least a note on this should be included, or better something like using dashed lines and different line widths can make it visible on the plot

Minor:
1.	Second line in Section 4. “(iii)” -> “(ii)”
2.	Last line in Section 6.2. “In shown in Table 2” - > “As shown in Table 2”

---

> ### Author Response · Authors · 2020-11-15
> **Response to AnonReviewer1**
>
> Thank you for your review. Your suggestions and comments are very helpful, and we have incorporated them into the paper. Here we would like to focus on your concerns and questions.
>
>
> (1.) “Are all components introduced really necessary? Why do we need to use DSCS-GP and not just use any function that grows cubically away from the data?”
>
> Technically speaking we can also achieve similar results with any function with super-quadratic growth away from the data. However, it is unclear whether they are principled and theoretically sound---they might work, but the resulting algorithm would be rather artificial and ad-hoc. RGPR, on the other hand, offers a well-justified function which arises from the observation that in GPs, we have infinitely many features that “cover” the whole input space, while in standard BNNs, we only use finitely many such features. By approaching our algorithm from this GP angle, we can arrive at a simple practical algorithm, as backed up by Prop. 1. Please refer also to our answer to AnonReviewer4 (point 3).
>
>
> (2.) “Overcomplicated method? Ablation study?”
>
> As shown in Algorithm 1 in the appendix, the resulting algorithm for RGPR is essentially what you have suggested: “Adding a cubic function to the pre-softmax units’ variance”. This is because we only need to compute $k_\leftrightarrow(x, x) = \frac{1}{N} \sum_{i=1}^N \frac{1}{3} |{x_i}|^3$ and subsequently add it to the pre-softmax’s variance for making predictions (for simplicity here we only consider the input $x$ and not other representations $h$ of $x$). This can be seen clearly from the definitions of $k_\leftrightarrow$ and $k^1$, and set $c_\text{min} = 0$ and $\bar{x} = \min(x, x) = x$. Note that in particular, we do not deal with GPs in practice---we do not even need to construct a kernel matrix. Using MC-integral, the only extra step we need to do during the prediction phase is to sample from a zero-mean Gaussian with variance $k_\leftrightarrow(x, x)$. For an ablation study on the effect of using hidden representations of $x$ instead of just cubic function in the input space, please refer to Fig. 3. In any case, we have updated the discussion about the practicality of RGPR in the last paragraph in Sec. 3.1. In particular we have moved Alg. 1 to the main text. Finally, even though RGPR algorithm is simple, this does not mean that our, admittedly technical, derivations are just a flourish. The value of this formalism is that it yields such a low-cost algorithm while simultaneously being motivated in a principled manner as approximating full non-parametric Bayesian inference, and associated with the theoretical guarantees we offer in the paper.
>
>
> (3.) Minor questions.
> * "Section 3.1. What about zeros?”: The origin has the kernel value $0$. We have clarified this in the text.
> * “What is n in the second paragraph in Section 6.1?”: It should be $N$---the dimensionality of the input space. Thank you for catching this.
> * “Figure 4.a – does this mean that all 3 lines coincide?”: Yes, all three lines coincide. We have made it clearer as suggested.

---

> > ### Comment · AnonReviewer1 · 2020-11-16
> > **Almost convincing arguments**
> >
> > Thank you for your answers. After reading all other reviews and your answers to that, you have made me convinced in the approach and I am raising the score.
> >
> > However, I believe you are still missing a "killer" argument for Bayesian sceptics. Personally, I like the beuty of how a GP connects as an infinite extension of a finite BNN in your work. But I do not think Bayesian sceptics and practicioners would be much impressed by this and some simple and undeniable argument is still missing...

---

### Author Response · Authors · 2020-11-20
**Thank you for your response!**

Dear AnonReviewer1, thank you for your quick (and positive) response! We really appreciate it.

AnonReviewer2 and AnonReviewer4, do our responses answer your questions/concerns? It would be very helpful for us to know your opinions so that we can address any follow-up issue before the discussion period ended.

Thank you for your cooperation!

The authors of paper #593

---

### Decision · Program_Chairs · 2021-01-07
**Final Decision**

**Decision:**

Reject

**Comment:**

The reviewers agree that the proposed method for reducing overconfidence in ReLU networks is novel and interesting. However, the presentation of the theoretical results is too informal and imprecise to warrant acceptance without a strong accompanying experimental section, which is unfortunately lacking. I therefore cannot recommend acceptance of the paper in its current form.